# WASSERSTEIN-BOUNDED GENERATIVE ADVERSARIAL NETWORKS

## ABSTRACT

In the field of Generative Adversarial Networks (GANs), how to design a stable training strategy remains an open problem. Wasserstein GANs have largely promoted the stability over the original GANs by introducing Wasserstein distance, but still remain unstable and are prone to a variety of failure modes. In this paper, we present a general framework named Wasserstein-Bounded GAN (WBGAN), which improves a large family of WGAN-based approaches by simply adding an upper-bound constraint to the Wasserstein term. Furthermore, we show that WBGAN can reasonably measure the difference of distributions which almost have no intersection. Experiments demonstrate that WBGAN can stabilize as well as accelerate convergence in the training processes of a series of WGAN-based variants.

## 1 INTRODUCTION

Over the past few years, Generative Adversarial Networks (GANs) have shown impressive results in many generative tasks. They are inspired by the game theory, that two models compete with each other: a generator which seeks to produce samples from the same distribution as the data, and a discriminator whose job is to distinguish between real and generated data. Both models are forced stronger simultaneously during the training process. GANs are capable of producing plausible synthetic data across a wide diversity of data modalities, including natural images (Karras et al., 2017; Brock et al., 2018; Lucic et al., 2019), natural language (Press et al., 2017; Lin et al., 2017; Rajeswar et al., 2017), music (Yang et al., 2017; Mogren, 2016; Dong et al., 2017; Dong & Yang, 2018), *etc.*

Despite their success, it is often difficult to train a GAN model in a fast and stable way, and researchers are facing issues like vanishing gradients, training instability, mode collapse, *etc*. This has led to a proliferation of works that focus on improving the quality of GANs by stabilizing the training procedure (Radford et al., 2015; Salimans et al., 2016; Zhao et al., 2016; Nowozin et al., 2016; Chen et al., 2016; Qi, 2017; Deshpande et al., 2018). In particular, Arjovsky et al. (2017) introduced a variant of GANs based on the Wasserstein distance, and releases the problem of gradient disappearance to some extent. However, WGANs limit the weight within a range to enforce the continuity of Lipschitz, which can easily cause over-simplified critic functions (Gulrajani et al., 2017). To solve this issue, Gulrajani et al. (2017) proposed a gradient penalty method termed WGAN-GP, which replaces the weight clipping in WGANs with a gradient penalty term. As such, WGAN-GP provides a more stable training procedure and succeeds in a variety of generating tasks. Based on WGAN-GP, more works (Wei et al., 2018; Petzka et al., 2017; Wu et al., 2018; Mescheder et al., 2018; Thanh-Tung et al., 2019; Kodali et al., 2017; Kim et al., 2018) adopt different forms of gradient penalty terms to further improve training stability. However, it is often observed that such gradient penalty strategy sometimes generate samples with unsatisfing quality, or even do not always converge to the equilibrium point (Mescheder et al., 2018).

In this paper, we propose a general framework named Wasserstein-Bounded GAN (WBGAN), which improve the stability of WGAN training by bounding the Wasserstein term. The highlight is that the instability of WGANs also resides in the dramatic changes of the estimated Wasserstein distance during the initial iterations. Many previous works just focused on improving the gradient penalty term for stable training, while they ignored the bottleneck of the Wasserstein term. The proposed training strategy is able to adaptively enforce the Wasserstein term within a certain value, so as to balance the Wasserstein loss and gradient penalty loss dynamically and make the training process more stable.

WBGANs are generalized, which can be instantiated using different kinds of bound estimations, and incorporated into any variant of WGANs to improve the training stability and accelerate the convergence. Specifically, with Sinkhorn distance (Cuturi, 2013; Genevay et al., 2017) for bound estimation, we test three representative variants of WGANs (WGAN-GP (Gulrajani et al., 2017), WGAN-div (Wu et al., 2018), and WGAN-GPReal (Mescheder et al., 2018)) on the CelebA dataset (Liu et al., 2015). As shown in Fig. 1, WBGANs outperform the corresponding counterparts, which demonstrates that the bounded strategy results in more stable training and accelerates the convergence.

## 2 BACKGROUNDS

**Wasserstein GANs (WGANs).** WGANs (Arjovsky et al., 2017) were primarily motivated by unstable training caused by the gradient vanishing problem of the original GANs (Goodfellow et al., 2014). They proposed to use 1-Wasserstein distance $W_1(\mathbb{P}_r, \mathbb{P}_g)$ to measure the difference between $\mathbb{P}_r$ and $\mathbb{P}_g$, the real and generated distributions, given that $W_1(\mathbb{P}_r, \mathbb{P}_g)$ is continuous everywhere and differentiable almost everywhere under mild assumptions. The objective of WGAN is formulated using the Kantorovich-Rubinstein duality (Villani, 2008):

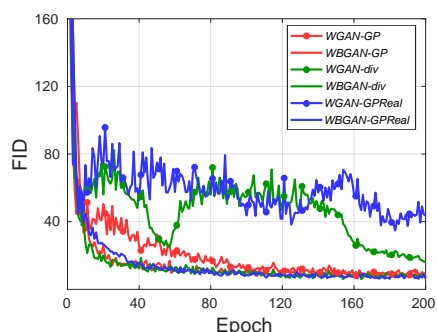

Figure 1: Instability in the training process of three variants of WGANs, *i.e.*, WGAN-GP, WGAN-div and WGAN-GPReal.

$$\min_G \max_{D \in \mathcal{L}_1} \mathbb{E}_{\boldsymbol{x} \sim \mathbb{P}_r}[D(\boldsymbol{x})] - \mathbb{E}_{\tilde{\boldsymbol{x}} \sim \mathbb{P}_g}[D(\tilde{\boldsymbol{x}})], \quad (1)$$

where $\mathcal{L}_1$ is the function space of all $D$ satisfying the 1-Lipschitz constraint $\|D\|_L \leq 1$. $D$ is a critic and $G$ is the generator, both of which are parameterized by a neural network. Under an optimal critic, minimizing the objective with respect to $G$ is to minimize $W_1(\mathbb{P}_r, \mathbb{P}_g)$. To enforce the 1-Lipschitz constraint on the critic, WGAN used a weight clipping on the critic to constrain the weights within a compact range, $[-c, c]$, which guarantees the set of critic functions is a subset of the $k$-Lipschitz functions for some $k$. With weight clipping, the critic tends to learn over-simplified functions (Gulrajani et al., 2017), which may lead to unsatisfying results. Gulrajani et al. (2017); Wei et al. (2018); Petzka et al. (2017); Wu et al. (2018) proposed different forms of gradient penalty as a regularization term, so that a generalized loss function with respect to the critic can be written as:

$$L_D = -[\mathbb{E}_{\boldsymbol{x} \sim \mathbb{P}_r}[D(\boldsymbol{x})] - \mathbb{E}_{\tilde{\boldsymbol{x}} \sim \mathbb{P}_g}[D(\tilde{\boldsymbol{x}})]] + \lambda_{\text{GP}} \cdot \text{GP}, \quad (2)$$

where $\mathbb{E}_{\boldsymbol{x} \sim \mathbb{P}_r}[D(\boldsymbol{x})] - \mathbb{E}_{\tilde{\boldsymbol{x}} \sim \mathbb{P}_g}[D(\tilde{\boldsymbol{x}})]$ stands for the Wasserstein term, and GP for the gradient penalty term. $L_D$ is actually posing a tradeoff between these two objectives.

**Wasserstein Distance between Empirical Distributions.** In practice, we approximate $W_1(\mathbb{P}_r, \mathbb{P}_g)$ using $W_1\left(\hat{\mathbb{P}}_r, \hat{\mathbb{P}}_g\right)$, where $\hat{\mathbb{P}}_r$ and $\hat{\mathbb{P}}_g$ denote the empirical version of $\mathbb{P}_r$ and $\mathbb{P}_g$ with $N$ samples, *i.e.*, $\hat{\mathbb{P}}_r = \frac{1}{N}\sum_{i=1}^{N} \delta_{\boldsymbol{y}_i}$, and $\hat{\mathbb{P}}_g = \frac{1}{N}\sum_{i=1}^{N} \delta_{G(\boldsymbol{z}_i)}$. Here, $\boldsymbol{y}_i$ is randomly sampled from the real image dataset, and $\delta_{\boldsymbol{y}_i}$ is the Dirac delta function at location $\boldsymbol{y}_i$. Computing $W_1\left(\hat{\mathbb{P}}_r, \hat{\mathbb{P}}_g\right)$ is a typical problem named discrete optimal transport. We denote $\mathcal{B}$ as the set of probabilistic couplings between two empirical distributions defined as:

$$\mathcal{B} := \{\boldsymbol{\Gamma} \in \mathbb{R}_+^{N \times N} \mid \boldsymbol{\Gamma} \mathbf{1}_N = \hat{\mathbb{P}}_g, \boldsymbol{\Gamma}^\top \mathbf{1}_N = \hat{\mathbb{P}}_r\}, \quad (3)$$

where $\mathbf{1}_N$ is a $N$-dimensional all-one vector. Then we have $W_1(\hat{\mathbb{P}}_r, \hat{\mathbb{P}}_g) = \min_{\gamma \in \mathcal{B}} \langle \boldsymbol{\Gamma}, \mathbf{C} \rangle_F$, where $\langle \cdot, \cdot \rangle_F$ is the Frobenius dot-product and $\mathbf{C}$ is the cost matrix, with each element $C_{i,j} = c(G(\boldsymbol{z}_i), \boldsymbol{y}_j)$ denoting the cost to move a probability mass from $G(\boldsymbol{z}_i)$ to $\boldsymbol{y}_j$. The optimal coupling is the solution of this minimization problem: $\boldsymbol{\Gamma}_0 = \arg\min_{\boldsymbol{\Gamma} \in \mathcal{B}} \langle \boldsymbol{\Gamma}, \mathbf{C} \rangle_F$.

**The Sinkhorn Algorithm.** Despite Wasserstein distance has appealing theoretical properties in measuring the difference between distributions, its computational costs for linear programming are often high in particular when the problem size becomes large. To alleviate this burden, Sinkhorn distance (Cuturi, 2013) was proposed to approximate Wasserstein distance:

$$d_\alpha(\hat{\mathbb{P}}_r, \hat{\mathbb{P}}_g) := \min_{P \in \mathcal{U}_\alpha(\hat{\mathbb{P}}_r, \hat{\mathbb{P}}_g)} \langle P, C \rangle, \quad (4)$$

where $\mathcal{U}_\alpha(\hat{\mathbb{P}}_r, \hat{\mathbb{P}}_g)$ is a subset of $\mathcal{B}$ defined in Eq. 3:

$$\mathcal{U}_\alpha(\hat{\mathbb{P}}_r, \hat{\mathbb{P}}_g) := \{\mathbf{\Gamma} \in \mathcal{B} | \mathbb{H}(\mathbf{\Gamma}) \geqslant \mathbb{H}(\hat{\mathbb{P}}_r) + \mathbb{H}(\hat{\mathbb{P}}_g) - \alpha\} \subset \mathcal{B}, \tag{5}$$

where $\mathbb{H}(\cdot)$ is the entropy defined as $\mathbb{H}(\mathbf{\Gamma}) = -\sum_{i,j=1}^N \Gamma_{i,j} \log \Gamma_{i,j}$ and $\mathbb{H}(\hat{\mathbb{P}}_r) = -\sum_{n=1}^N \hat{p}_n \log \hat{p}_n$ where $\hat{p}_n$ is the probability of the $n$-th sample. Compared to Wasserstein distance, Sinkhorn distance restricts the search space of joint probabilities to those with sufficient smoothness. To compute Sinkhorn distance, a Lagrange multiplier was used:

$$d^\lambda(\hat{\mathbb{P}}_r, \hat{\mathbb{P}}_g) = \langle \mathbf{\Gamma}^\lambda, \mathbf{C} \rangle, \quad \mathbf{\Gamma}^\lambda = \arg\min_{\mathbf{\Gamma} \in \mathcal{B}} \langle \mathbf{\Gamma}, \mathbf{C} \rangle - \frac{1}{\lambda} \mathbb{H}(\mathbf{\Gamma}). \tag{6}$$

Each $\alpha$ corresponds a $\lambda \in [0, \infty)$ such that $d_\alpha(\hat{\mathbb{P}}_r, \hat{\mathbb{P}}_g) = d^\lambda(\hat{\mathbb{P}}_r, \hat{\mathbb{P}}_g)$ holds for that pair $(\hat{\mathbb{P}}_r, \hat{\mathbb{P}}_g)$. $d^\lambda(\cdot, \cdot)$ can be computed with a much cheaper cost than the original Wasserstein distance using matrix scaling algorithms. For $\lambda > 0$, the solution $\mathbf{\Gamma}^\lambda$ is unique and has the form $\mathbf{\Gamma}^\lambda = \mathrm{diag}(u)\mathbf{K}\,\mathrm{diag}(v)$, where $\mathbf{K}$ is the element-wise exponential of $-\lambda\mathbf{C}$. $u$ and $v$ are two non-negative vectors uniquely defined up to a multiplicative factor (Cuturi, 2013).

# 3 WASSERSTEIN-BOUNDED GANS

## 3.1 BOUND CONSTRAINT ON 1-WASSERSTEIN DISTANCE

We start with Eq. 2 and denote $W = \mathbb{E}_{\boldsymbol{x} \sim \mathbb{P}_r}[D(\boldsymbol{x})] - \mathbb{E}_{\tilde{\boldsymbol{x}} \sim \mathbb{P}_g}[D(\tilde{\boldsymbol{x}})]$. $W$ is often referred to as the Wasserstein term, which is unbounded during the training process. In a wide range of WGAN's variants such as WGAN-GP (Gulrajani et al., 2017), the critic defined by $L_D$ is to maximize the Wasserstein term $W$ while satisfying the gradient penalty GP. However, in practice, we find that $W$ often rises rapidly to a tremendous value which is far from rational during the initial training procedure. A possible reason may lie in that the critic function does not satisfy the Lipschitz constraint during the initial training stage. As shown in Fig. 2, this leads to dramatic instability in optimization and finally results in unsatisfying performance in image generation.

Our idea is thus straightforward, *i.e.*, setting an upper-bound for $W$. The modified critic loss function is written as:
$$L_D = -W + [W - \overline{W}]_+ + \lambda_{\mathrm{GP}} \cdot \mathrm{GP}, \tag{7}$$
where $[\cdot]_+ = \max\{\cdot, 0\}$ is the ramp function that ignores negative inputs, and $\overline{W}$ denotes the upper bound of W, and will be discussed later. If $W \leqslant \overline{W}$, the term $[W - \overline{W}]_+$ simply vanishes and Eq. 7 is equivalent to Eq. 2; otherwise, we have $-W + [W - \overline{W}]_+ = -\overline{W}$, which implies that the modified Wasserstein term is bounded by $\overline{W}$.

Our formulation brings a benefit to the numerical stability of the Wasserstein term. In practice, it remains comparable to the other term, $\lambda_{\mathrm{GP}} \cdot \mathrm{GP}$, so that both $W$ and GP can be optimized in a 'mild' manner, *i.e.*, without any one of them dominating or being ignored during training. Note that the $\overline{W}$ term cannot be chosen arbitrarily. Setting it too small, $\overline{W}$ will limit the capacity of the critic function, resulting in a poor generation. Setting it too large, there will be no effect of bounding the $W$ term.

The proposed bounded strategy is a general framework. We name it general in two folds: First, WBGAN can be applied to almost all gradient penalty based WGANs, such as WGAN-GP (Gulrajani et al., 2017), WGAN-GPReal (Mescheder et al., 2018), *etc*. Moreover, there are different ways to estimate the value of $\overline{W}$. For example, the linear programming was applied successfully to some existing WGANs like WGAN-TS (Liu et al., 2018). In what follows, we present an example which uses Sinkhorn distance to estimate $\overline{W}$, while we believe other ways of estimation are also possible.

## 3.2 WBGAN WITH SINKHORN DISTANCE

In this section, we give an instantiation, Sikhorn distance (Cuturi, 2013), to effectively compute the bounded term $\overline{W}$. The motivation of using Sinkhorn distance lies in that in theory, the Wasserstein term of WGAN will eventually converge to the 1-Wasserstein distance between the real distribution $\mathbb{P}_r$ and the generated distribution $\mathbb{P}_g$ (Arjovsky et al., 2017; Gulrajani et al., 2017). Therefore, we can use the 1-Wasserstein distance between the empirical distributions, $\hat{\mathbb{P}}_r$ and $\hat{\mathbb{P}}_g$, as the upper-bound $\overline{W}$. Since the computation of Wasserstein distance involves a large linear programming which

---

**Algorithm 1** WBGAN with Sinkhorn distance

---

**Require:** learning rate $\alpha$, batch size $M$, the number of iterations of the critic per generator iteration $N_{\text{critic}}$, weight of gradient penalty $\lambda_{\text{GP}}$, weight of Sinkhorn distance $\lambda_{\text{s}}$, initial parameters $\theta$ and $\phi_0$, other hyper-parameters;
1: **while** $\phi_t$ has not converged **do**
2:     **for** $n = 1, \ldots, N_{\text{critic}}$ **do**
3:         Sample a batch $\{\boldsymbol{x}^{(m)}\}_{m=1}^{M} \sim \mathbb{P}_r$ from real data;
4:         Sample a batch $\{\boldsymbol{z}^{(m)}\}_{m=1}^{M} \sim \mathbb{P}_{\boldsymbol{z}}$ of prior samples;
5:         $W \leftarrow \frac{1}{M}\sum_{m=1}^{M} D_\theta(\boldsymbol{x}^{(m)}) - \frac{1}{M}\sum_{m=1}^{M} D_\theta(G_{\phi_t}(\boldsymbol{z}^{(m)}))$;
6:         Calculate Sinkhorn distance $d^\lambda(\hat{\mathbb{P}}_r, \hat{\mathbb{P}}_g)$ between $\{\boldsymbol{x}^{(m)}\}_{m=1}^{M}$ and $\{G_{\phi_t}(\boldsymbol{z}^{(m)})\}_{m=1}^{M}$;
7:         $L_\theta \leftarrow -W + [W - d^\lambda(\hat{\mathbb{P}}_r, \hat{\mathbb{P}}_g)]_+ + \lambda_{\text{GP}} \cdot \text{GP}$;
8:         $\theta \leftarrow \text{Adam}(L_\theta, \theta, \alpha, \beta_1, \beta_2)$;
9:     **end for**
10:    Sample a batch $\{\boldsymbol{z}^{(m)}\}_{m=1}^{M} \sim \mathbb{P}_{\boldsymbol{z}}$ of prior samples;
11:    Calculate Sinkhorn distance $d^\lambda(\hat{\mathbb{P}}_r, \hat{\mathbb{P}}_g)$ between $\{\boldsymbol{x}^{(m)}\}_{m=1}^{M}$ and $\{G_{\phi_t}(\boldsymbol{z}^{(m)})\}_{m=1}^{M}$;
12:    $L_{\phi_t} \leftarrow -\mathbb{E}_{\boldsymbol{z} \sim \mathbb{P}_{\boldsymbol{z}}}[D_\theta(G_{\phi_t}(\boldsymbol{z}))] + \lambda_{\text{s}} \cdot d^\lambda(\hat{\mathbb{P}}_r, \hat{\mathbb{P}}_g)$;
13:    $\phi_{t+1} \leftarrow \text{Adam}(L_{\phi_t}, \phi_t, \alpha, \beta_1, \beta_2)$;
14: **end while**
**Ensure:** trained parameters $\theta$ and $\phi_T$ (converged).

---

suffers heavy computational costs, we replace it by Sinkhorn distance instead – the Sinkhorn distance between $\hat{\mathbb{P}}_r$ and $\hat{\mathbb{P}}_g$ can be computed using Sinkhorn's matrix scaling algorithm (Cuturi, 2013), which is orders of magnitude faster than the linear programming solvers.

Mathematically, consider a generator function $G_\phi(\boldsymbol{z})$ that produces samples by transforming noise input $\boldsymbol{z}$ drawn from a simple distribution $\mathbb{P}_{\boldsymbol{z}}$, *e.g.*, Gaussian distribution. $D_\theta$ stands for a critic function parameterized by $\theta$. The objective of the critic is:

$$
\begin{aligned}
L_\theta(\mathbb{P}_r, \mathbb{P}_g) = \max_{D_\theta \in \mathcal{L}_1} \quad & \mathbb{E}_{\boldsymbol{x} \sim \mathbb{P}_r}[D_\theta(\boldsymbol{x})] - \mathbb{E}_{\boldsymbol{x} \sim \mathbb{P}_g}[D_\theta(\boldsymbol{x})] \\
& - \Big[\mathbb{E}_{\boldsymbol{x} \sim \mathbb{P}_r}[D_\theta(\boldsymbol{x})] - \mathbb{E}_{\boldsymbol{x} \sim \mathbb{P}_g}[D_\theta(\boldsymbol{x})] - d^\lambda(\hat{\mathbb{P}}_r, \hat{\mathbb{P}}_g)\Big]_+ ,
\end{aligned}
\tag{8}
$$

where $d^\lambda(\hat{\mathbb{P}}_r, \hat{\mathbb{P}}_g)$ is the Sinkhorn distance defined in Eq. 6. On the other hand, given a fixed critic function $D_{\theta^\star}$, considering that Sinkhorn distance allows gradient back-propagation (Genevay et al., 2017), we can find the optimal generator $G_{\phi^\star}$ by solving:

$$
\phi^\star = \arg\min_\phi \quad -\mathbb{E}_{\boldsymbol{z} \sim \mathbb{P}_{\boldsymbol{z}}}[D_{\theta^\star}(G_\phi(\boldsymbol{z}))] + \lambda_{\text{s}} \cdot d^\lambda(\hat{\mathbb{P}}_r, \hat{\mathbb{P}}_g),
\tag{9}
$$

where $\lambda_{\text{s}}$ is a balancing hyper-parameter, which we set $\lambda_{\text{s}} = 0.5$ in this paper. In Algorithm 1, we summarize the flowchart of training WBGAN with Sinkhorn distance.

### 3.2.1 RELATIONSHIP BETWEEN $d^\lambda(\hat{\mathbb{P}}_r, \hat{\mathbb{P}}_g)$ AND $W_1(\mathbb{P}_r, \mathbb{P}_g)$

We employ $d^\lambda(\hat{\mathbb{P}}_r, \hat{\mathbb{P}}_g)$ as an approximation of 1-Wasserstein distance $W_1(\mathbb{P}_r, \mathbb{P}_g)$. Let $(X, d)$ be a separable metric space. $\mathcal{P}(X)$ denotes the set of Borel probability measures. $\mathcal{P}_p(X)$ denotes the set of all $\mu \in \mathcal{P}(X)$ such that $\int_X d(\boldsymbol{x}, \boldsymbol{y})^p d\mu(\boldsymbol{x}) < +\infty$ for some $\boldsymbol{y} \in X$. We can suppose real data distribution $\mathbb{P}_r$, generated data distribution $\mathbb{P}_g$ and their empirical distribution $\hat{\mathbb{P}}_r$ and $\hat{\mathbb{P}}_g$ all in $\mathcal{P}_p(X)$.

**Proposition 1.** Let $\mathbb{P}_r$ and $\mathbb{P}_g$ be real data distribution and generated data distribution. Suppose that $\hat{\mathbb{P}}_r$ and $\hat{\mathbb{P}}_g$ are empirical measures of $\mathbb{P}_r$ and $\mathbb{P}_g$. Then we have $0 \leqslant W_1(\mathbb{P}_r, \mathbb{P}_g) \leqslant \mathbb{E}[W_1(\hat{\mathbb{P}}_r, \hat{\mathbb{P}}_g)]$.

*Proof.* Please refer to Appendix A. $\qquad\square$

Proposition 1 tells us that as $\mathbb{E}[W_1(\hat{\mathbb{P}}_r, \hat{\mathbb{P}}_g)] \to 0$, $W_1(\mathbb{P}_r, \mathbb{P}_g)$ is forced to 0. Cuturi (2013) has pointed out that if $\lambda$ is chosen large enough, $d^\lambda(\hat{\mathbb{P}}_r, \hat{\mathbb{P}}_g)$ coincides with $W_1(\hat{\mathbb{P}}_r, \hat{\mathbb{P}}_g)$. So, it is reasonable to use $d^\lambda(\hat{\mathbb{P}}_r, \hat{\mathbb{P}}_g)$ to constrain the Wasserstein term.

## 3.3 ANALYSIS OF WBGAN

Most GANs measure the distance between distributions based on probability divergence. We will prove that the Eq. 8 is indeed a valid divergence. First, we have the following definition.

**Definition 1.** Given probability measures $p$ and $q$, $\mathcal{D}$ is a functional of $p$ and $q$. If $\mathcal{D}$ satisfies the following properties:

$$
\begin{aligned}
&1. \quad \mathcal{D}(p, q) \geq 0; \\
&2. \quad p = q \iff \mathcal{D}(p, q) = 0,
\end{aligned}
\tag{10}
$$

then we say $\mathcal{D}$ is a probability divergence between $p$ and $q$.

**Remark 1.** The following $W(\mathbb{P}_r, \mathbb{P}_g)$ satisfies the Definition 1 and is therefore a probability divergence.

$$
W(\mathbb{P}_r, \mathbb{P}_g) = \max_{D \in \mathcal{L}_1} \mathbb{E}_{\boldsymbol{x} \sim \mathbb{P}_r}[D(\boldsymbol{x})] - \mathbb{E}_{\tilde{\boldsymbol{x}} \sim \mathbb{P}_g}[D(\tilde{\boldsymbol{x}})],
\tag{11}
$$

where $\mathcal{L}_1$ is the 1-Lipschitz constraint. Please see the proof and detailed discussion in Su (2018). This is the objective of critic used by WGAN (Arjovsky et al., 2017).

**Remark 2.** Equation 8 satisfies the Definition 1 and is a probability divergence.

*Proof.* The proof is given in Appendix B. □

**Remark 3.** Consider two distributions $\mathbb{P}_r(\boldsymbol{x}) = \delta(\boldsymbol{x} - \boldsymbol{\alpha})$, $\mathbb{P}_g(\boldsymbol{x}) = \delta(\boldsymbol{x} - \boldsymbol{\beta})$ that have no intersection ($\boldsymbol{\alpha} \neq \boldsymbol{\beta}$). $\delta$ is the Dirac delta function. In such an extreme case, Eq. 8 can still be optimized by gradient descent.

*Proof.* The proof is in Appendix C □

Remark 2 tells us that Eq. 8 is a valid divergence. Since the real data distribution is supported by low-dimensional manifolds, the supports of generated distribution and real data distribution are unlikely to have a non-negligible intersection. Remark 3 shows that compared to the standard GAN (Goodfellow et al., 2014), WBGAN can continuously measure the difference between two distributions, even if there is almost no intersection between the distributions.

# 4 EXPERIMENTS

## 4.1 SETTINGS AND BASELINES

To verify that WBGAN is a generalized approach, we select three variants of WGAN, namely, WGAN-GP (Gulrajani et al., 2017), WGAN-div (Wu et al., 2018) and WGAN-GPReal (gradient penalty on real data only) (Mescheder et al., 2018) as our baselines. By adding bound constraints to these WGAN variants, we obtain the counterparts WBGAN-GP, WBGAN-div, and WBGAN-GPReal, respectively. Two different network architectures are used, *i.e.,* DCGAN (Radford et al., 2015) and BigGAN (Brock et al., 2018). For DCGAN, we directly output the activation before the sigmoid layer. BigGAN is a conditional GAN (Mirza & Osindero, 2014) architecture, in which class conditioning is passed to generator by supplying it with class-conditional gains and biases in the batch normalization layer (Ioffe & Szegedy, 2015; de Vries et al., 2017; Dumoulin et al., 2017). In addition, the discriminator is conditioned (Miyato & Koyama, 2018) by using the cosine similarity between its features and a set of learned class embedding. We use the spectral norm (Miyato et al., 2018) in BigGAN, but for the sake of simplicity, we do not use the self-attention module (Wang et al., 2017; Zhang et al., 2018). Other hyper-parameters and the network architecture of BigGAN simply follow the original paper.

We choose the Fréchet Inception Distance (FID) (Heusel et al., 2017) for quantitative evaluation, which has been proven to be more consistent with individual assessment in evaluating the fidelity and variation of the generated image samples.

## 4.2 MID-RESOLUTION EXPERIMENTS

We first investigate mid-resolution image generation on the CelebA dataset (Liu et al., 2015), a large-scale face image dataset with more than 200K face images. During training, we crop $108 \times 108$ face from the original images and then resize them to $64 \times 64$.

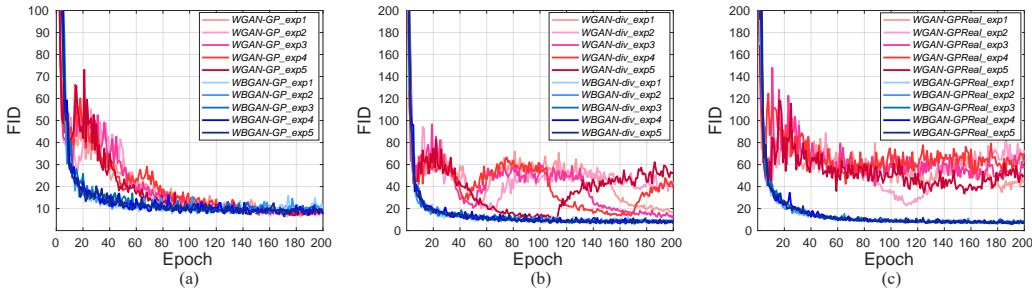

Figure 2: FID curves on the CelebA dataset, with WGAN-GP, WGAN-div and WGAN-GPReal as baselines, respectively. Each figure contains 5 individual runs for both each counterpart.

Table 1: FID comparison between WGAN-based methods and WBGAN-based methods. The BigGAN architecture uses spectral normalization in the generator and discriminator, and the number of conditional labels is set to be 1 because the training dataset only contains face images.

| Network architecture | Loss | Dataset | Resolution | Batch | G Param(M) | D Param(M) | FID |
|---|---|---|---|---|---|---|---|
| DCGAN | WGAN-GP | CelebA | $64 \times 64$ | 128 | 5.1 | 4.3 | $6.76 \pm 0.17$ |
| | WBGAN-GP (ours) | | | | | | $7.32 \pm 0.55$ |
| | WGAN-div | | | | | | $13.94 \pm 2.67$ |
| | WBGAN-div (ours) | | | | | | $6.26 \pm 0.30$ |
| | WGAN-GPReal | | | | | | $34.92 \pm 6.84$ |
| | WBGAN-GPReal (ours) | | | | | | $\mathbf{6.01 \pm 0.33}$ |
| BigGAN | WGAN-GP | CelebA | $64 \times 64$ | 128 | 8.4 | 4.9 | 13.39 |
| | WBGAN-GP (ours) | | | | | | $\mathbf{6.97}$ |
| | WGAN-div | | | | | | 45.93 |
| | WBGAN-div (ours) | | | | | | 7.23 |
| | WGAN-GPReal | | | | | | 42.71 |
| | WBGAN-GPReal (ours) | | | | | | 9.61 |

**FID Stability.** We first use DCGAN to build our generator and discriminator. Training curves are shown in Fig. 2, and quantitative results are summarized in Table 1. Each approach is executed for 5 times and the average is reported. All FID curves are obtained from generators directly without using the moving average strategy (Karras et al., 2017; Mescheder et al., 2018; Brock et al., 2018; Yazıcıet al., 2018) to avoid over-smoothing the FID curves, such that we can diagnose the underlying oscillating properties of different methods during training. One can see that WBGAN-based counterparts improve the stability during training, and achieve superior performance over the WGAN-based baselines. We emphasize that the converged FID values reported by WBGAN-div and WBGAN-GPReal are lower than those reported by WGAN-div and WGAN-GPReal. In particular, WGAN-div suffers several FID fluctuation unexpectedly, and WGAN-GPReal has not ever achieved FID convergence during the entire training process. Regarding WGAN-GP, although the final FID is slightly better than that of WBGAN-GP (6.76 vs. 7.32), we observe a much slower convergence rate in Fig. 2(a). For the generated face images by different approaches, please refer to Fig. 10 in Appendix F for details.

We also investigate a stronger backbone by replacing the network with BigGAN, a conditional GAN architecture that uses spectral normalization on both generator and discriminator. We set the number of labels to be 1 since the CelebA dataset only contains face images. Training curves are shown in Fig. 3 and quantitative results are summarized in Table 1. Among three WGAN-based methods, only WGAN-GP achieves convergence, but its convergence speed and the FID value are inferior to those reported by WBGAN-GP. In opposite, both WGAN-div and WGAN-GPReal fails to converge while the counterparts equipped with WBGAN perform well. For the generated face images by different approaches, please refer to Fig. 11 and Fig. 12 in Appendix F for details.

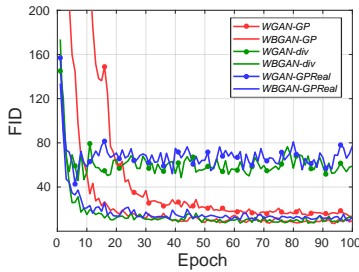

Figure 3: FID curves of BigGAN-based approaches on the CelebA dataset.

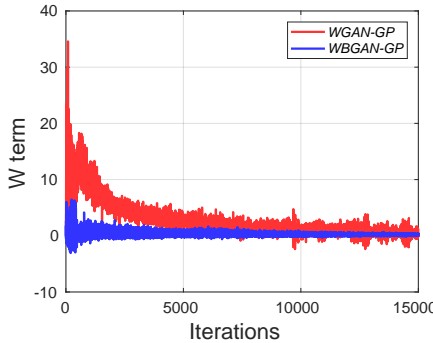

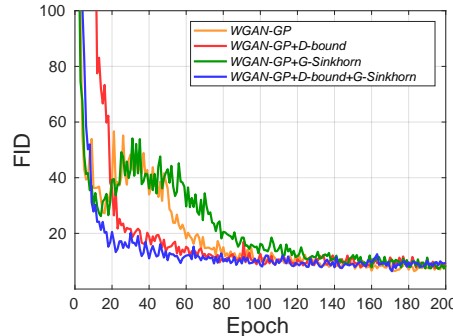

Figure 4: Curves of the Wasserstain term, produced by WGAN-GP and WBGAN-GP.

Figure 5: FID curves of DCGAN-based ablation study on the CelebA dataset.

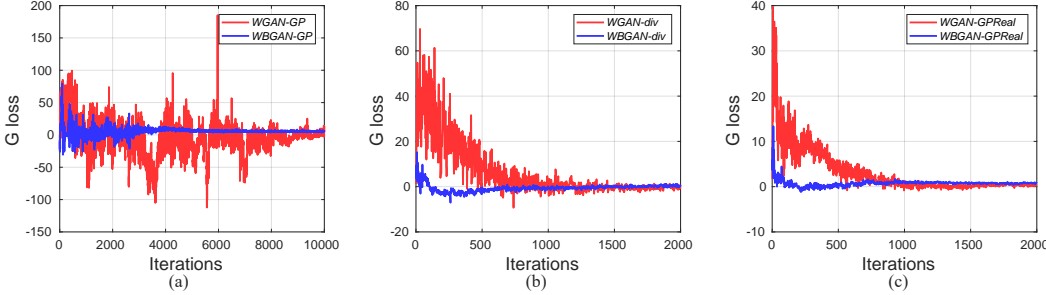

Figure 6: Generator loss in the beginning iterations. BigGAN on CelebA. (a) WGAN-GP vs WBGAN-GP, (b) WGAN-div vs WBGAN-div, (c) WGAN-GPReal vs WBGAN-GPReal.

**Wasserstein Loss and Generator Loss Stability.** Next, we evaluate the stability of WBGAN in terms of the Wasserstein term and generator loss. In Fig. 4, we evaluate the impact on WGAN-GP (DCGAN on CelebA). One can see that, after the bound is applied, the Wasserstein term $W$ is stablized especially during the start of training. Due to space limit, more results using BigGAN on CelebA are provided in Appendix E. In addition, we compute a new term named the generator loss, which is defined as $G_{\text{loss}} = -\mathbb{E}_{\boldsymbol{z} \sim \mathbb{P}_{\boldsymbol{z}}}[D_\theta(G_\phi(\boldsymbol{z}))]$. Fig. 6 shows the curves of this statistics during the starting iterations. Compared to WGAN-based approaches, WBGAN-based approaches produce more stable $G_{\text{loss}}$ terms, which verifies that the training process of GAN becomes more stable.

**Ablation Study.** Before continuing to high-resolution experiments, we conduct an ablation study to investigate the contribution made by different components of WBGAN. The backbone network is DCGAN, and the dataset is CelebA. We compare four configurations, *i.e.*, WGAN-GP, with the original loss term used in WGAN-GP; WGAN-GP+$D$-bound, which adds a bound (Sinkhorn distance) to the Wasserstein term of the critic $D$ of WGAN-GP; WGAN-GP+$G$-Sinkhorn, which adds Sinkhorn distance to the loss function of the generator $G$ in WGAN-GP; and WGAN-GP+$D$-bound+$G$-Sinkhorn, which is equivalent to the final WBGAN-GP, with Sinkhorn distance added to both critic $D$ and generator $G$. Fig. 5 plots the FID curves of all four settings. One can see that, although the FID curves of WGAN-GP and WGAN-GP+$G$-Sinkhorn descend quickly in the first 10 epochs, they begin to fluctuate between 20 to 40 epochs. On the other hand, when WGAN-GP is combined with $D$-bound, FID is able to descend smoothly (without fluctuation), showing that it is the bounded constraint that stablizes the training process. Finally, by integrating both $D$-bound and $G$-Sinkhorn into WGAN-GP, the FID curve descends not only smoothly but also fast, which is what we desire in real-world applications.

### 4.3 HIGH-RESOLUTION EXPERIMENTS AND REMARKS

In this section, we evaluate our approach on higher-resolution ($128 \times 128$) images. We use the CelebA-HQ dataset (Karras et al., 2017), and use BigGAN (Brock et al., 2018) as the backbone. As

Table 2: FID comparison in high-dimensional experiments.

| Network architecture | Loss | Dataset | Resolution | Batch | G Param(M) | D Param(M) | FID |
|---|---|---|---|---|---|---|---|
| BigGAN | WGAN-GP | CelebA-HQ | $128 \times 128$ | 64 | 8.4 | 9.6 | 17.58 |
| | WBGAN-GP (ours) | | | | | | 18.32 |
| | WGAN-div | | | | | | 21.05 |
| | WBGAN-div (ours) | | | | | | 17.26 |
| | WGAN-GPReal | | | | | | 21.33 |
| | WBGAN-GPReal (ours) | | | | | | **12.87** |

Figure 7: Curves of FID using BigGAN network architecture on the CelebA-HQ dataset. (a) WGAN-GP vs WBGAN-GP, (b) WGAN-div vs WBGAN-div, (c) WGAN-GPReal vs WBGAN-GPReal. Lower is better.

the target become larger ($128 \times 128$), the number of images we can feed into a single batch becomes smaller (64). Since we are using an empirical way of estimating Sinkhorn distance, it becomes less accurate in the scenario of small batch size and large image size. In other words, it is no longer the best choice to use Sinkhorn distance to estimate the upper-bound $\overline{W}$.

Returning to our generalized formulation, Eq. 7, we note that other forms of bound to constrain the critic. Here we consider a very simple bound, which is also based on empirical study. Note that the baseline methods, though not converging very well, can finally arrive at a stablized $W$ value. Heuristically, we use this constant value (there is no need to be accurate) as the bound, which is 10 for WGAN-GP, 5 for WGAN-div and 3 for WGAN-GPReal, respectively. In Appendix D, we provide the curves of the Wasserstein term for these baselines, which lead to our estimation.

FID curves and quantitative results using these constant bounds are shown in Fig. 7 and Table 2, respectively. We find that WBGAN-GP produces a similar convergence rate with WGAN-GP, WBGAN-div is slightly better than WGAN-div, and WBGAN-GPReal outperforms WGAN-GPReal and produces the best results. For the generated face images by different approaches, please refer to Fig. 13 and Fig. 14 in Appendix F for details.

**Discussions.** From the above experiments, we can see that Sinkhorn distance is just one way of upper-bound estimation. In case that it becomes less accurate, we can freely replace it with other types of estimation. Besides the constant bound used above, there also exist other examples, such as the two-step computation of the exact Wasserstein distance (Liu et al., 2018). However, it is still a challenge to estimate the Wasserstein distance between high-resolution ($1024 \times 1024$) image distributions efficiently. Nevertheless, the most important deliveries of our work are that a bounded Wasserstein term can bring benefits on training stability, and that we can use it to a wide range of frameworks based on WGAN.

## 5 CONCLUSIONS

This paper introduced a general framework called WBGANs, which can be applied to a variety of WGAN variants to stabilize the training process and improve the performance. We clarify that WBGANs can stabilize the Wasserstein term at the beginning of the iterations, which is beneficial for smoother convergence of WGAN-based methods. We present an instantiated bound estimation method via Sinkhorn distance and give a theoretical analysis on it. It remains an open topic on how to set a better bound for higher resolution image generation tasks.

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

## A    PROOF OF PROPOSITION 1

*Proof.* Suppose $\mu, \upsilon_1, \upsilon_2 \in \mathcal{P}_p(X)$, $t_1, t_2 \geq 0$, $t_1 + t_2 = 1$, then there exist $\gamma_1(x, y)$ and $\gamma_2(x, y)$ with marginals $(\mu, \upsilon_1)$ and $(\mu, \upsilon_2)$ satisfying:

$$W_1(\mu, \upsilon_1) = \int_{X \times X} \|x - y\|_1 d\gamma_1(x, y), \tag{12}$$

$$W_1(\mu, \upsilon_2) = \int_{X \times X} \|x - y\|_1 d\gamma_2(x, y). \tag{13}$$

Let $\upsilon = t_1\upsilon_1 + t_2\upsilon_2$, $\gamma(x, y) = t_1\gamma_1(x, y) + t_2\gamma_2(x, y)$, then $\gamma(x, y)$ has marginals $(\mu, \upsilon)$. We can derive:

$$
\begin{aligned}
W_1(\mu, \upsilon) &\leq \int_{X \times X} \|x - y\|_1 d\gamma(x, y) \\
&= t_1 \int_{X \times X} \|x - y\|_1 d\gamma_1(x, y) + t_2 \int_{X \times X} \|x - y\|_1 d\gamma_2(x, y) \\
&= t_1 W_1(\mu, \upsilon_1) + t_2 W_1(\mu, \upsilon_2).
\end{aligned}
\tag{14}
$$

This conclusion can be extended to a general form:

$$W_1(\mu, \upsilon) \leq t_1 W_1(\mu, \upsilon_1) + t_2 W_1(\mu, \upsilon_2) + \ldots t_n W_1(\mu, \upsilon_n), \tag{15}$$

where $\upsilon_1, \upsilon_2, \ldots \upsilon_n \in \mathcal{P}_p(x)$, $t_1, t_2, \ldots t_n \geq 0$, $t_1 + t_2 + \cdots + t_n = 1$, $\upsilon = t_1\upsilon_1 + t_2\upsilon_2 + \cdots + t_n\upsilon_n$. Suppose $\hat{\mathbb{P}}_{g_i}(1 \leq i \leq n)$ are the independent empirical measures drawn from $\hat{\mathbb{P}}_g$. From Eq. 15, we can get

$$W_1\left(\mathbb{P}_r, \frac{1}{n}\sum_{i=1}^{n}\hat{\mathbb{P}}_{g_i}\right) \leq \frac{1}{n}\sum_{i=1}^{n}W_1(\mathbb{P}_r, \hat{\mathbb{P}}_{g_i}). \tag{16}$$

According to the strong law of large numbers, we can derive that with probability 1, $\frac{1}{n}\sum_{i=1}^{n}\hat{\mathbb{P}}_{g_i} \to \mathbb{P}_g$ as $n \to \infty$ (assuming $\mathbb{P}_g$ has finite first moments). Since $W_1$ is continuous in $\mathcal{P}_p(X)$, we can derive that with probability 1, $W_1(\mathbb{P}_r, \frac{1}{n}\sum_{i=1}^{n}\hat{\mathbb{P}}_{g_i}) \to W_1(\mathbb{P}_r, \mathbb{P}_g)$ as $n \to \infty$. By the law of large numbers again, with probability 1, $\frac{1}{n}\sum_{i=1}^{n}W_1(\mathbb{P}_r, \hat{\mathbb{P}}_{g_i}) \to \mathbb{E}[W_1(\mathbb{P}_r, \hat{\mathbb{P}}_g)]$ as $n \to \infty$. Thus we can deduce that:

$$W_1(\mathbb{P}_r, \mathbb{P}_g) \leq \mathbb{E}[W_1(\mathbb{P}_r, \hat{\mathbb{P}}_g)]. \tag{17}$$

Similarly, suppose $\hat{\mathbb{P}}_{r_i}(1 \leq i \leq n)$ are the independent empirical measures drawn from $\hat{\mathbb{P}}_r$. Since the symmetry of Wasserstein distance, we can deduce that:

$$W_1(\mathbb{P}_r, \hat{\mathbb{P}}_g) \leq \mathbb{E}[W_1(\hat{\mathbb{P}}_r, \hat{\mathbb{P}}_g)]. \tag{18}$$

Therefore, combining Eq. 17 and Eq. 18, we can get $W_1(\mathbb{P}_r, \mathbb{P}_g) \leq \mathbb{E}[W_1(\hat{\mathbb{P}}_r, \hat{\mathbb{P}}_g)]$.

## B    PROOF OF REMARK 2

*Proof.* First, let $D_\theta(\boldsymbol{x}) \equiv 0$, then

$$
\begin{aligned}
L_\theta(\mathbb{P}_r, \mathbb{P}_g) = \max_{D_\theta \in \mathcal{L}_1} \quad & \mathbb{E}_{\boldsymbol{x} \sim \mathbb{P}_r}[D_\theta(\boldsymbol{x})] - \mathbb{E}_{\boldsymbol{x} \sim \mathbb{P}_g}[D_\theta(\boldsymbol{x})] \\
& - \left[\mathbb{E}_{\boldsymbol{x} \sim \mathbb{P}_r}[D_\theta(\boldsymbol{x})] - \mathbb{E}_{\boldsymbol{x} \sim \mathbb{P}_g}[D_\theta(\boldsymbol{x})] - d^\lambda(\hat{\mathbb{P}}_r, \hat{\mathbb{P}}_g)\right]_+ \\
\geq \; & \mathbb{E}_{\boldsymbol{x} \sim \mathbb{P}_r}[0] - \mathbb{E}_{\boldsymbol{x} \sim \mathbb{P}_g}[0] - \left[\mathbb{E}_{\boldsymbol{x} \sim \mathbb{P}_r}[0] - \mathbb{E}_{\boldsymbol{x} \sim \mathbb{P}_g}[0] - d^\lambda(\hat{\mathbb{P}}_r, \hat{\mathbb{P}}_g)\right]_+ \\
= \; & 0,
\end{aligned}
\tag{19}
$$

where $d^\lambda(\hat{\mathbb{P}}_r, \hat{\mathbb{P}}_g) \geq 0$ is the Sinkhorn distance defined in Eq. 6.

Next, if $\mathbb{P}_r = \mathbb{P}_g$, then we have $L_\theta(\mathbb{P}_r, \mathbb{P}_g) = 0$. So we only need to show $L_\theta(\mathbb{P}_r, \mathbb{P}_g) > 0$ if $\mathbb{P}_r \neq \mathbb{P}_g$.

Let $D_\theta(\boldsymbol{x}) = sign(\mathbb{P}_r(\boldsymbol{x}) - \mathbb{P}_g(\boldsymbol{x}))$, we have

$$
\begin{aligned}
w &= \mathbb{E}_{\boldsymbol{x} \sim \mathbb{P}_r}[D_\theta(\boldsymbol{x})] - \mathbb{E}_{\boldsymbol{x} \sim \mathbb{P}_g}[D_\theta(\boldsymbol{x})] \\
&= \int (\mathbb{P}_r(\boldsymbol{x}) - \mathbb{P}_g(\boldsymbol{x})) \cdot sign(\mathbb{P}_r(\boldsymbol{x}) - \mathbb{P}_g(\boldsymbol{x}))dx \\
&> 0
\end{aligned}
\tag{20}
$$

Applying this into Eq. 8 leads to

$$
\begin{aligned}
L_\theta(\mathbb{P}_r, \mathbb{P}_g) &= \max_{D_\theta \in \mathcal{L}_1} \quad \mathbb{E}_{\boldsymbol{x} \sim \mathbb{P}_r}[D_\theta(\boldsymbol{x})] - \mathbb{E}_{\boldsymbol{x} \sim \mathbb{P}_g}[D_\theta(\boldsymbol{x})] \\
&\qquad - \left[ \mathbb{E}_{\boldsymbol{x} \sim \mathbb{P}_r}[D_\theta(\boldsymbol{x})] - \mathbb{E}_{\boldsymbol{x} \sim \mathbb{P}_g}[D_\theta(\boldsymbol{x})] - d^\lambda(\hat{\mathbb{P}}_r, \hat{\mathbb{P}}_g) \right]_+ \\
&\geq w - \left[ w - d^\lambda(\hat{\mathbb{P}}_r, \hat{\mathbb{P}}_g) \right]_+ \\
&= \begin{cases} w, & \text{if } w \leq d^\lambda(\hat{\mathbb{P}}_r, \hat{\mathbb{P}}_g) \\ d^\lambda(\hat{\mathbb{P}}_r, \hat{\mathbb{P}}_g). & \text{otherwise} \end{cases}
\end{aligned}
\tag{21}
$$

Since $\mathbb{P}_r \neq \mathbb{P}_g$, we know that $d^\lambda(\hat{\mathbb{P}}_r, \hat{\mathbb{P}}_g) > 0$. Therefore, we have $L_\theta(\mathbb{P}_r, \mathbb{P}_g) > 0$ while $\mathbb{P}_r \neq \mathbb{P}_g$. We finish the proof.

## C  PROOF OF REMARK 3

*Proof.* Let $\mathbb{P}_r(\boldsymbol{x}) = \delta(\boldsymbol{x} - \boldsymbol{\alpha})$, $\mathbb{P}_g(\boldsymbol{x}) = \delta(\boldsymbol{x} - \boldsymbol{\beta})$ and $\boldsymbol{\alpha} \neq \boldsymbol{\beta}$, then we have

$$
\begin{aligned}
L_\theta(\mathbb{P}_r, \mathbb{P}_g) &= \max_{D_\theta \in \mathcal{L}_1} \quad \mathbb{E}_{\boldsymbol{x} \sim \mathbb{P}_r}[D_\theta(\boldsymbol{x})] - \mathbb{E}_{\boldsymbol{x} \sim \mathbb{P}_g}[D_\theta(\boldsymbol{x})] \\
&\qquad - \left[ \mathbb{E}_{\boldsymbol{x} \sim \mathbb{P}_r}[D_\theta(\boldsymbol{x})] - \mathbb{E}_{\boldsymbol{x} \sim \mathbb{P}_g}[D_\theta(\boldsymbol{x})] - d^\lambda(\hat{\mathbb{P}}_r, \hat{\mathbb{P}}_g) \right]_+ \\
&= \max_{D_\theta \in \mathcal{L}_1} \quad D_\theta(\boldsymbol{\alpha}) - D_\theta(\boldsymbol{\beta}) - \left[ D_\theta(\boldsymbol{\alpha}) - D_\theta(\boldsymbol{\beta}) - d^\lambda(\hat{\mathbb{P}}_r, \hat{\mathbb{P}}_g) \right]_+ \\
&= \max_{D_\theta \in \mathcal{L}_1} \begin{cases} D_\theta(\boldsymbol{\alpha}) - D_\theta(\boldsymbol{\beta}), & \text{if } D_\theta(\boldsymbol{\alpha}) - D_\theta(\boldsymbol{\beta}) \leq d^\lambda(\hat{\mathbb{P}}_r, \hat{\mathbb{P}}_g) \\ d^\lambda(\hat{\mathbb{P}}_r, \hat{\mathbb{P}}_g). & \text{otherwise} \end{cases}
\end{aligned}
\tag{22}
$$

We know that Wasserstein distance $W(\mathbb{P}_r, \mathbb{P}_g) = \max_{D_\theta \in \mathcal{L}_1} D_\theta(\boldsymbol{\alpha}) - D_\theta(\boldsymbol{\beta})$. Since $\mathbb{P}_r, \mathbb{P}_g$ are Dirac distributions, then we have $W(\mathbb{P}_r, \mathbb{P}_g) = d^\lambda(\hat{\mathbb{P}}_r, \hat{\mathbb{P}}_g)$. Combining this into Eq. 22 leads to $L_\theta(\mathbb{P}_r, \mathbb{P}_g) = d^\lambda(\hat{\mathbb{P}}_r, \hat{\mathbb{P}}_g)$.

Considering that Sinkhorn distance $d^\lambda(\hat{\mathbb{P}}_r, \hat{\mathbb{P}}_g)$ (Cuturi, 2013) allows gradient back-propagation, we finish the proof.

## D  CURVES OF THE WASSERSTEIN TERM ON CELEBA-HQ DATASET

Fig. 8 shows the convergence curves of Wasserstein term for three WGAN methods. Convergence values are different for different WGANs. For example, WGAN-GP converges to 10. WGAN-div is 5. WGAN-GPReal is 3. We use these values as bound.

## E  ADDITIONAL STABILITY EXPERIMENTS ON THE WASSERSTEIN TERM

Fig. 9 shows the curves of Wasserstein term in the beginning iterations. Compared to the WGAN-based method, WBGAN-based method is more stable. The network architecture is BigGAN and the dataset is CelebA.

## F  SAMPLES AND INTERPOLATIONS FROM FACE MODELS

Here we display a few generated samples of face images by different approaches on the CelebA and CelebA-HQ datasets.

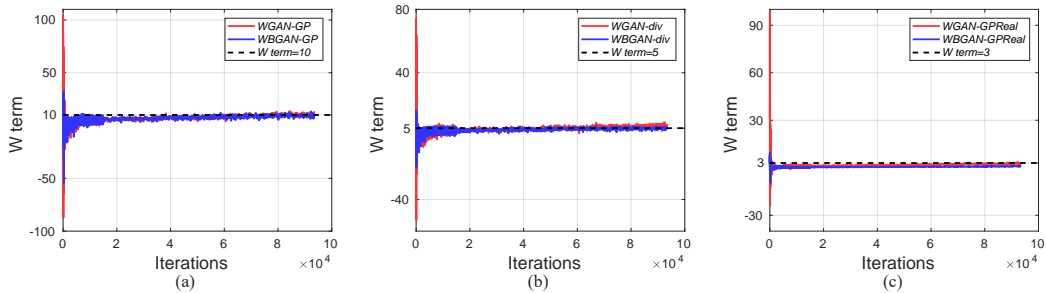

Figure 8: Wasserstein term for all training iterations. (a) WGAN-GP vs WBGAN-GP, (b) WGAN-div vs WBGAN-div, (c) WGAN-GPReal vs WBGAN-GPReal.

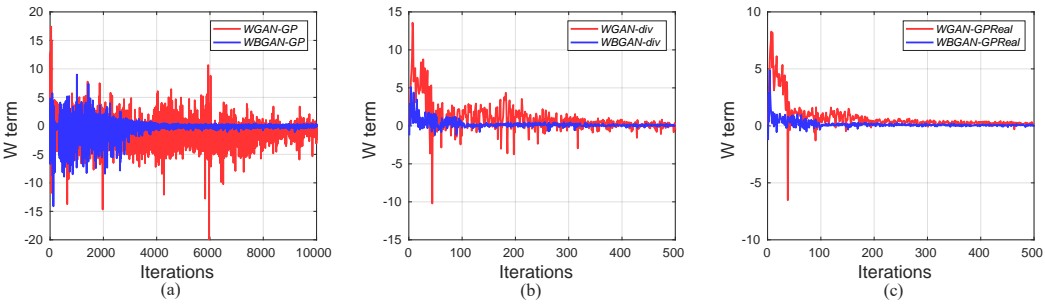

Figure 9: Wasserstein term in the beginning iterations. (a) WGAN-GP vs WBGAN-GP, (b) WGAN-div vs WBGAN-div, (c) WGAN-GPReal vs WBGAN-GPReal.

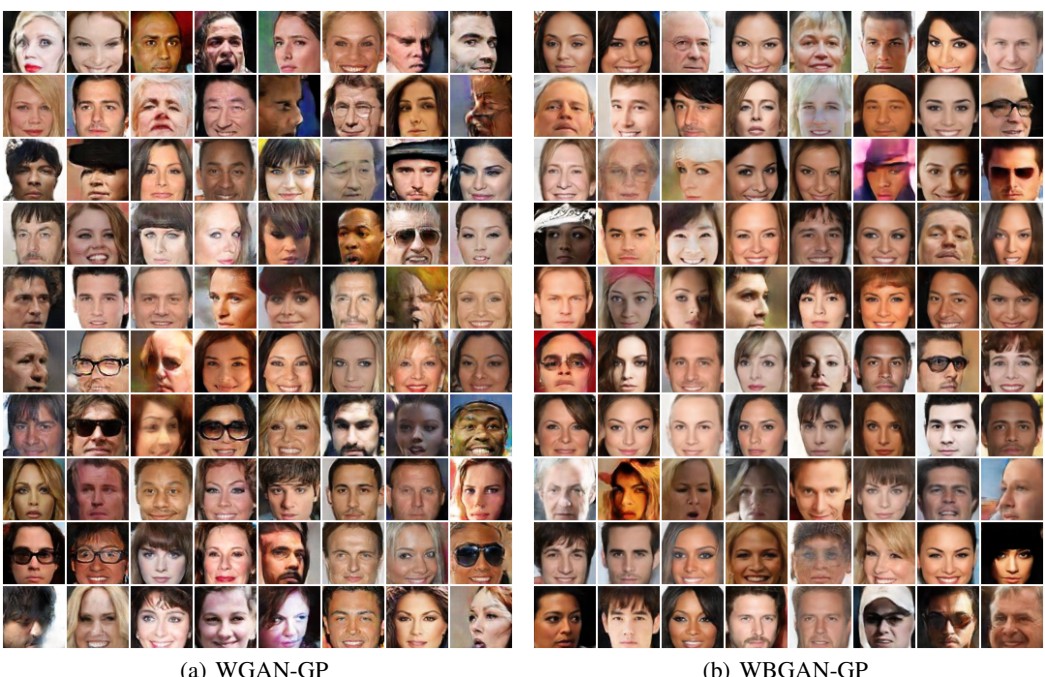

(a) WGAN-GP       (b) WBGAN-GP

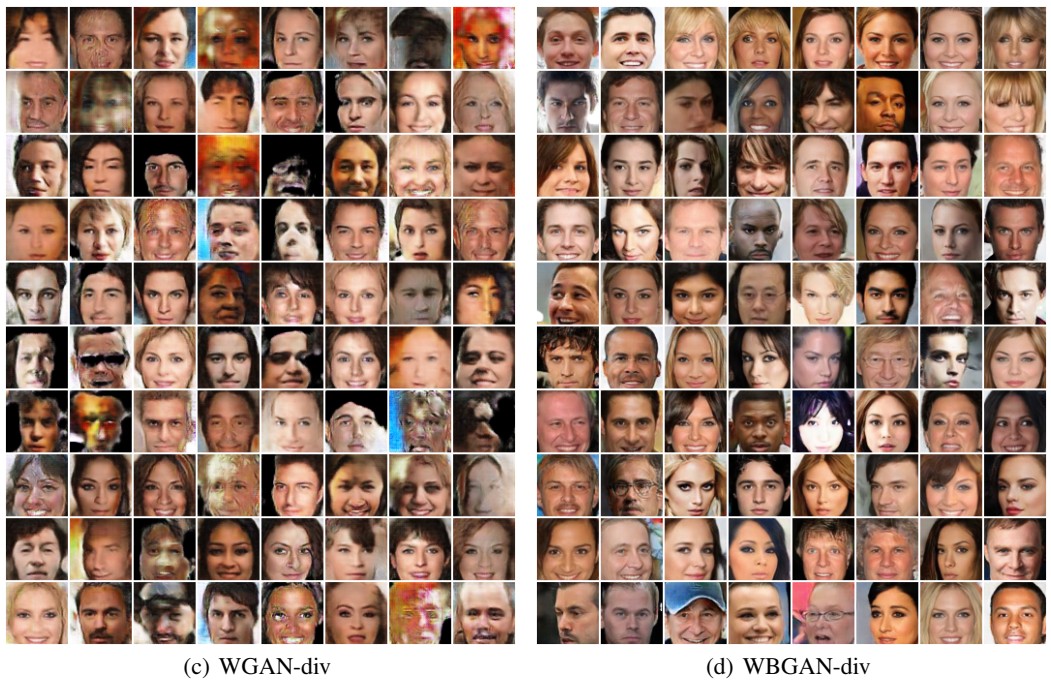

(c) WGAN-div

(d) WBGAN-div

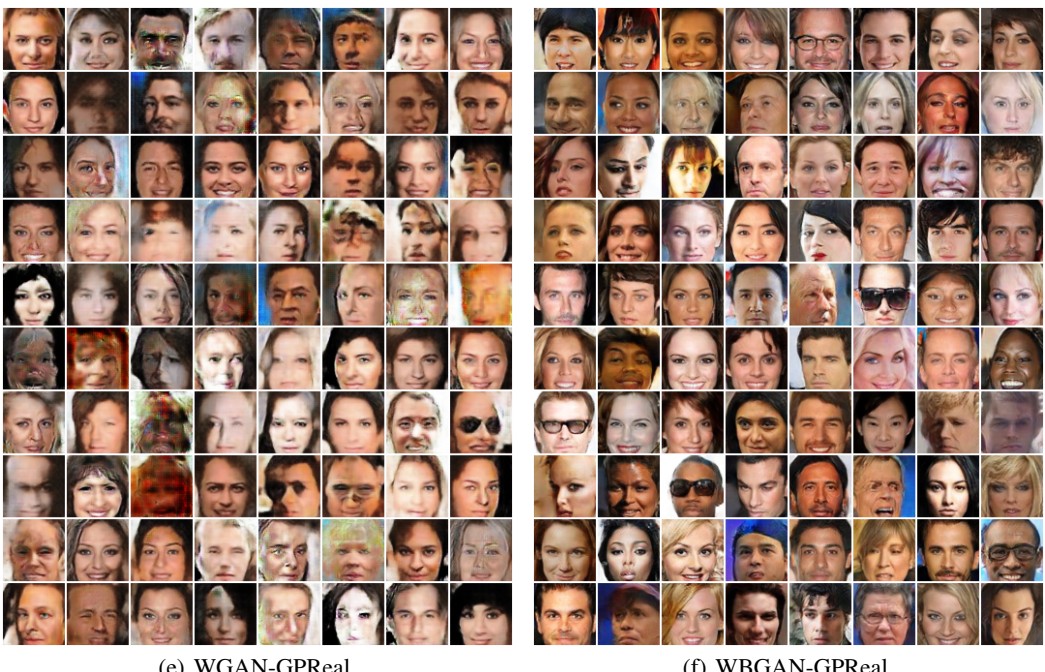

(e) WGAN-GPReal

(f) WBGAN-GPReal

Figure 10: Samples of DCGAN on CelebA64

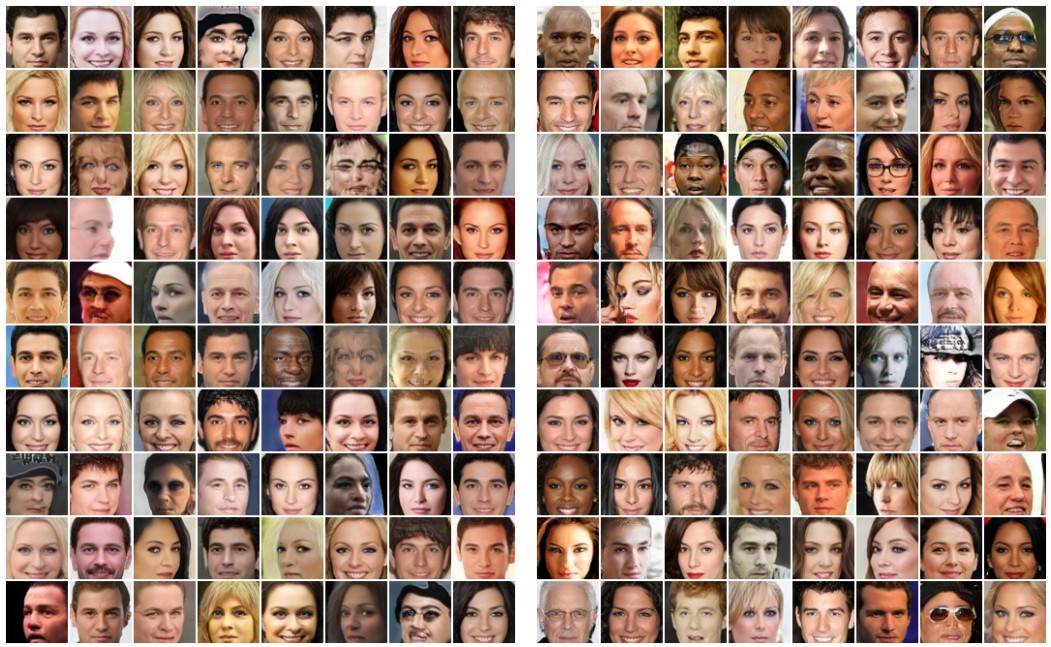

(a) WGAN-GP            (b) WBGAN-GP

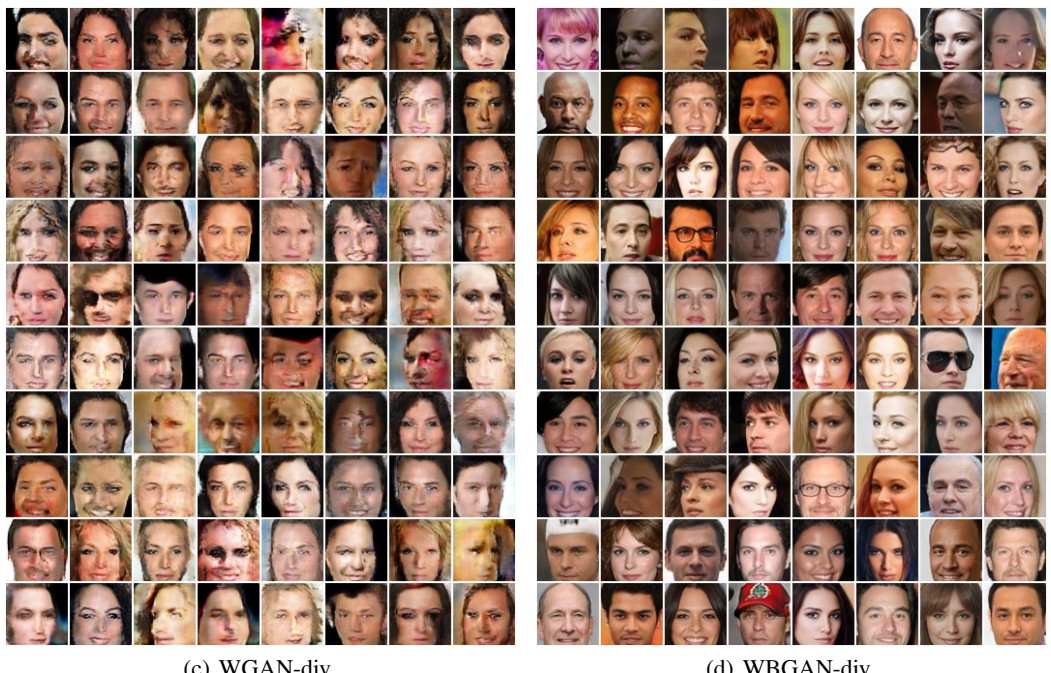

(c) WGAN-div            (d) WBGAN-div

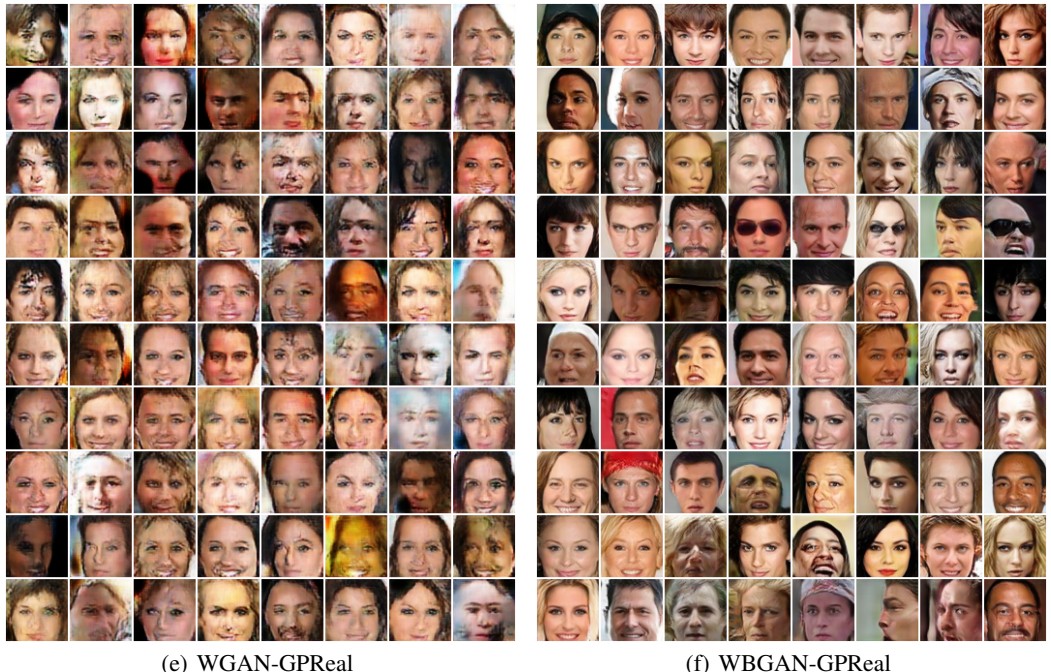

(e) WGAN-GPReal                                    (f) WBGAN-GPReal

Figure 11: Samples of BigGAN on CelebA64

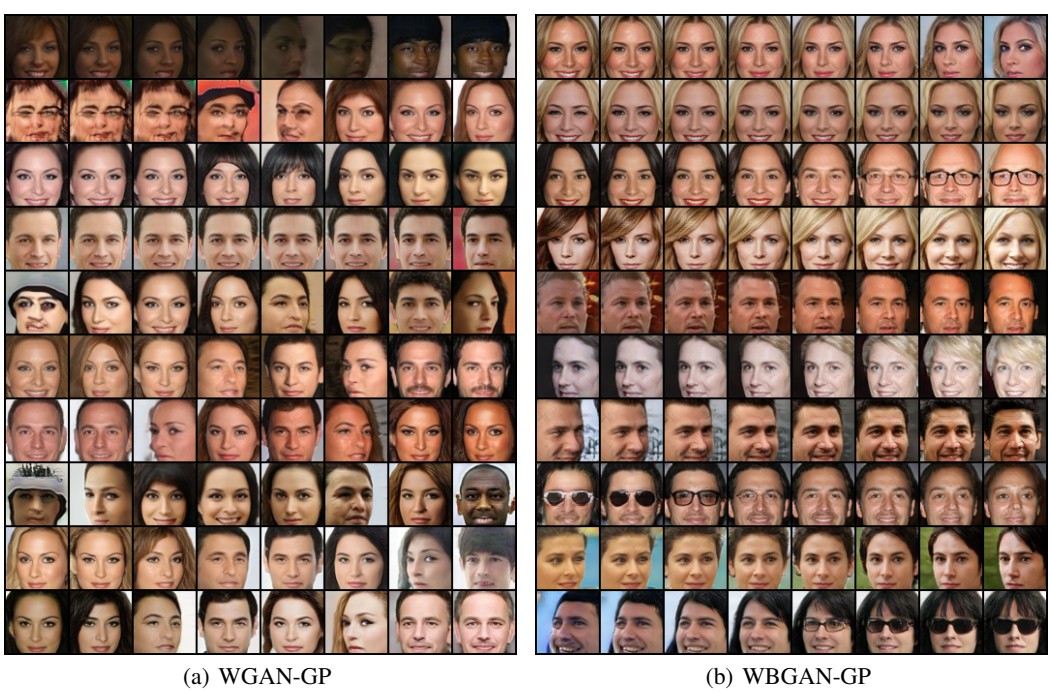

(a) WGAN-GP                                         (b) WBGAN-GP

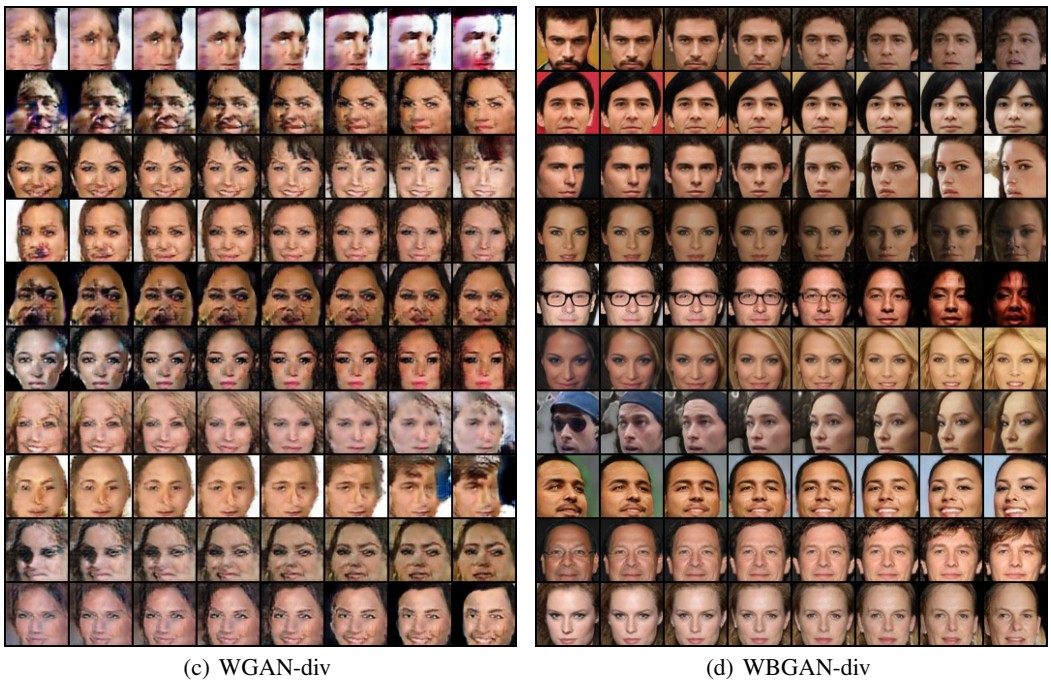

(c) WGAN-div          (d) WBGAN-div

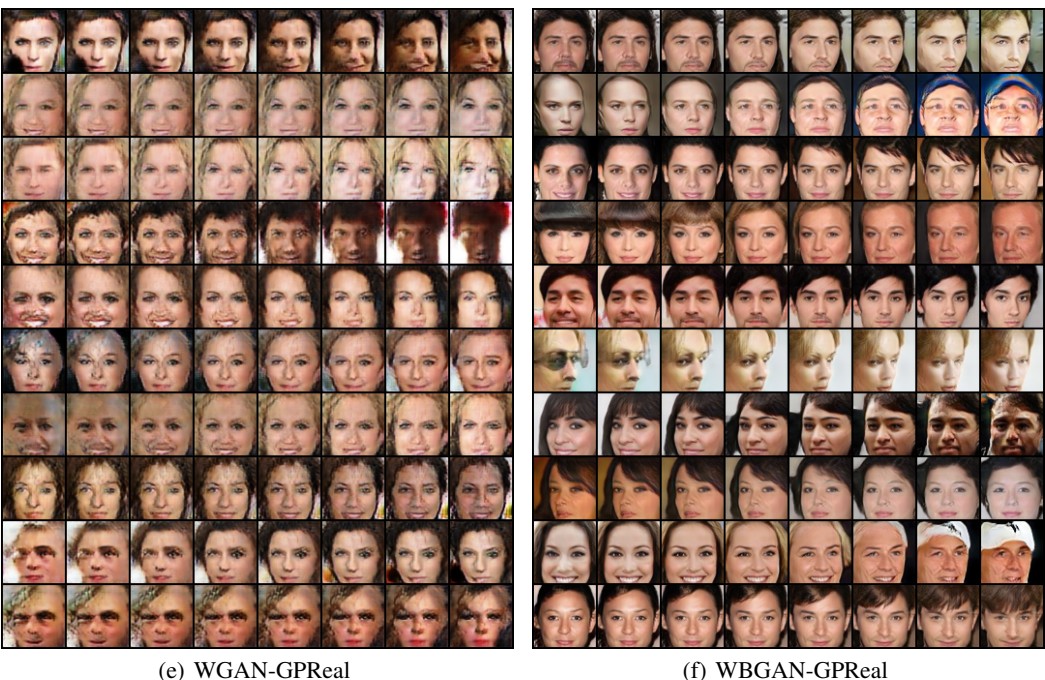

(e) WGAN-GPReal          (f) WBGAN-GPReal

Figure 12: Interpolations of BigGAN between $z$ on CelebA64

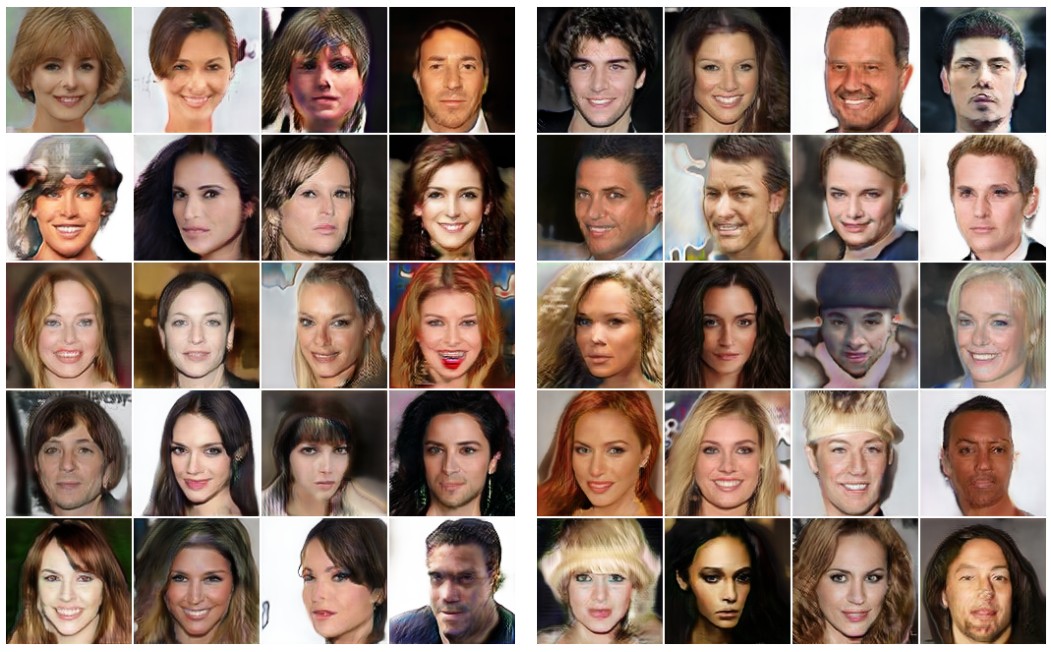

(a) WGAN-GP                              (b) WBGAN-GP

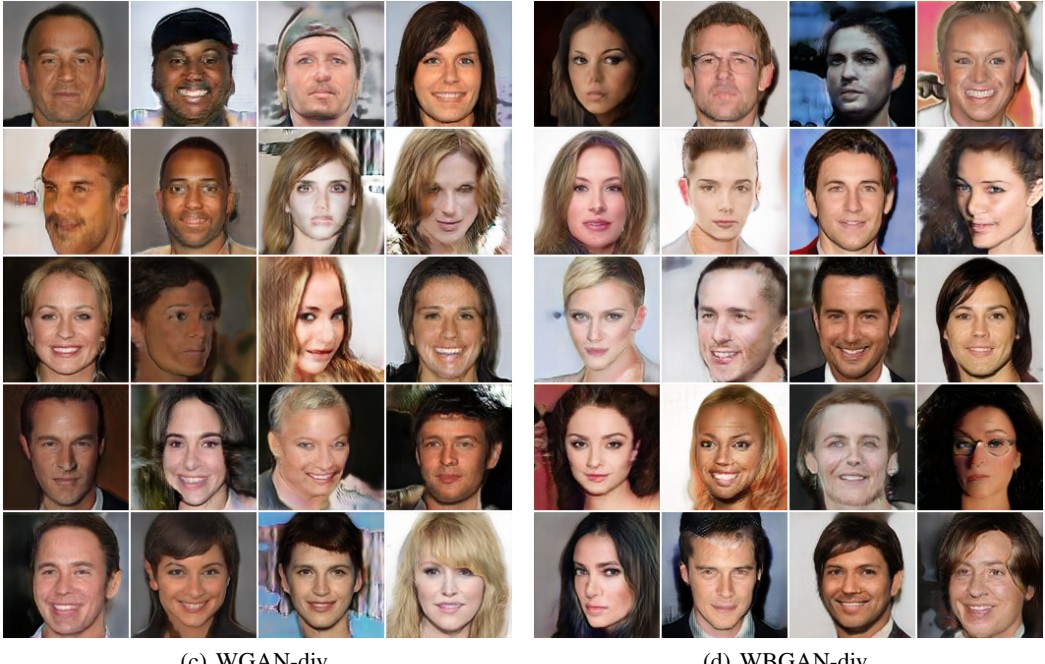

(c) WGAN-div                             (d) WBGAN-div

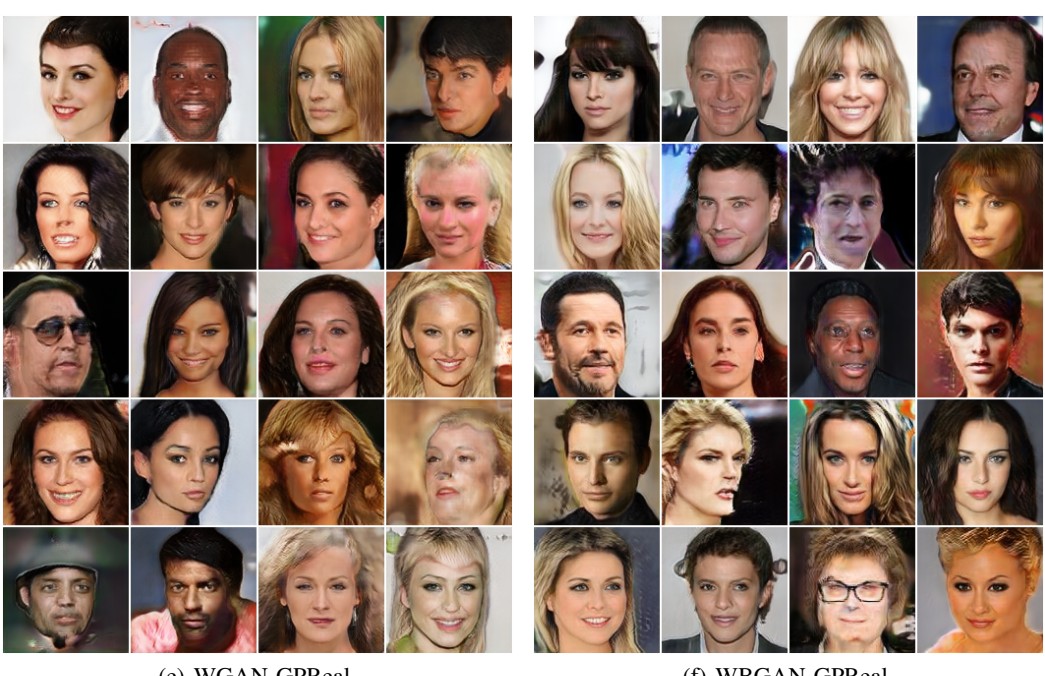

(e) WGAN-GPReal        (f) WBGAN-GPReal

Figure 13: Samples of BigGAN on CelebA-HQ128

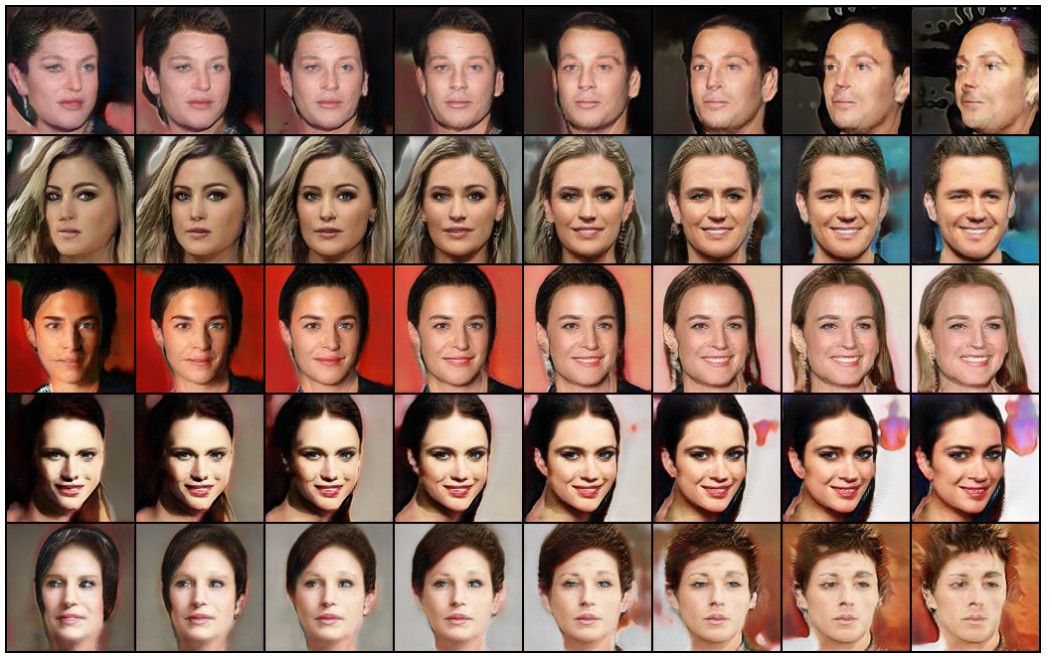

(a) WGAN-GP

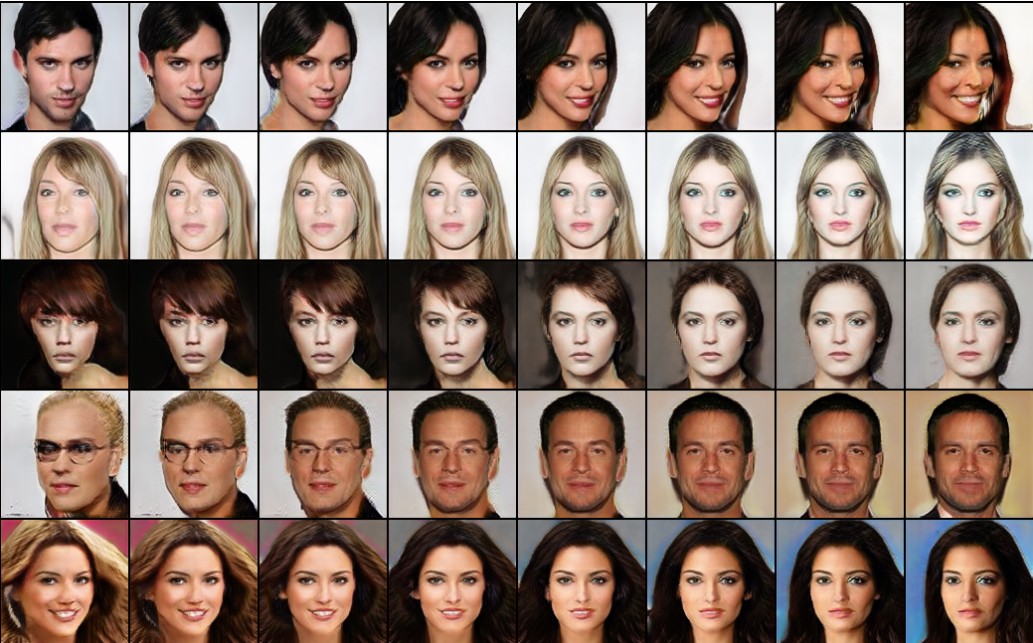

(b) WBGAN-GP

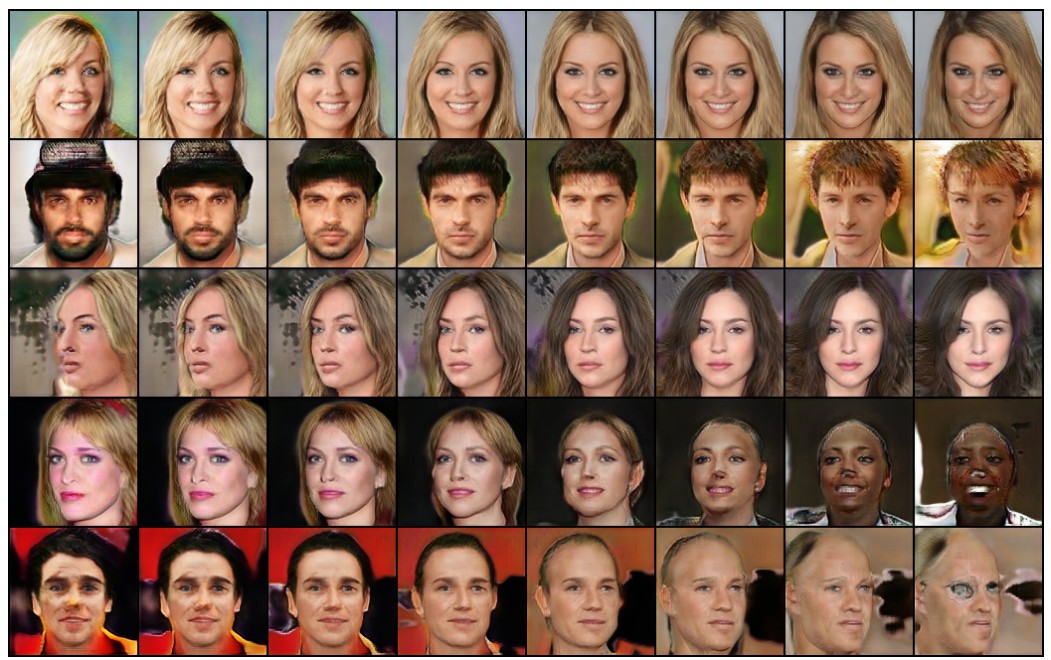

(c) WGAN-div

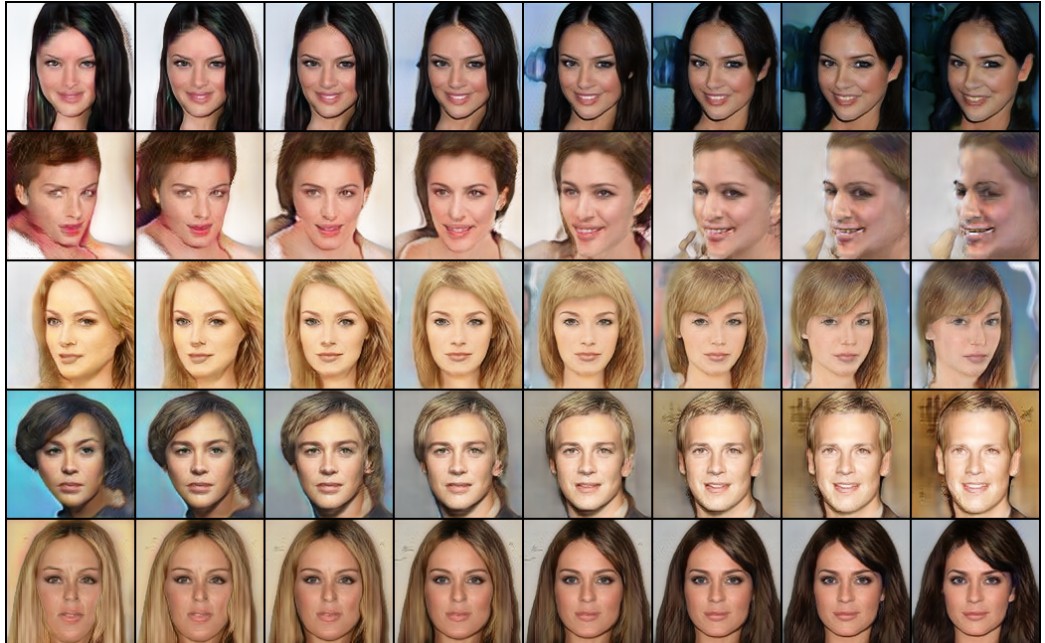

(d) WBGAN-div

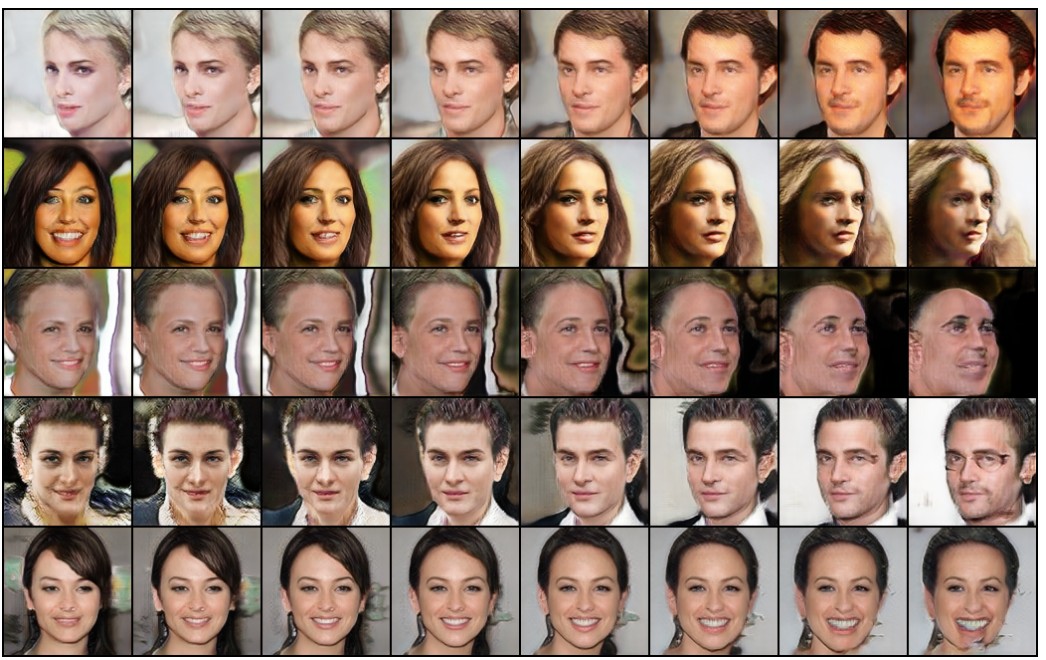

(e) WGAN-GPReal

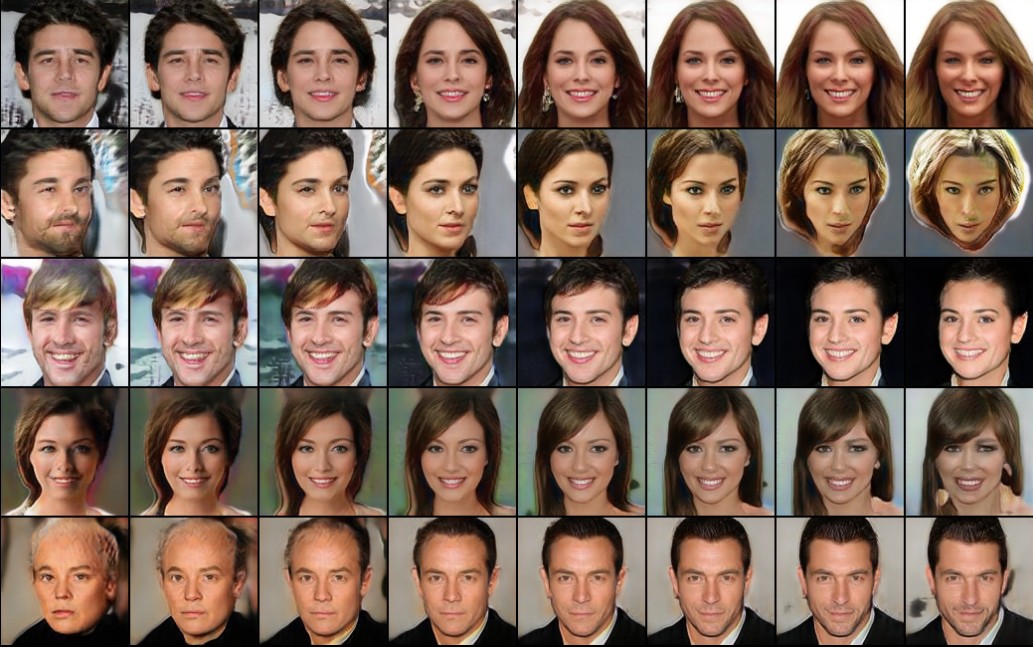

(f) WBGAN-GPReal

Figure 14: Interpolations of BigGAN between $z$ on CelebA-HQ128

