# OpenReview forum: "Wasserstein-Bounded Generative Adversarial Networks"
_ICLR.cc/2020/Conference — Reject_

### Official Review · AnonReviewer1 · 2019-10-17
**Official Blind Review #1**

**Rating:** 1

**Review:**

The paper proposes a new way of stabilizing Wasserstein GANs by using Sinkhorn distance to upper-bound the objective of WGAN's critic's loss during the training. GAN stabilization is a well-motivated problem and limiting the dramatic changes of discriminator loss clearly helps achieving this goal. Experiments show that in few settings the proposed method successfully addresses this issue. However, the theoretical insights have multiple flaws and incorrect proofs, while the choice of experiments raises few questions. Finally, the resolution limitations suggest the overall contribution is rather incremental.

The paper thus requires further work in terms of design, theory and experiments; for instance the choice of Sinkhorn distance as a boundary-heuristic might be too limiting. Although the main bounding strategy seems to be a promising idea, this work does not meet the quality requirements of ICLR.

Pros:
A. Conceptual simplicity of the general method of bounding discriminator's objective.
B. The method works well for a few variants of Wasserstein GAN on medium resolutions (64x64, 128x128).

Issues:
1. Resolution limitation. The upper-bound cannot be reasonably computed for higher resolutions in mini-batch training.
2. It is unclear how costly the method is in terms of resources and training time.
3. Experiments were done with CelebA/CelebA-hq with DCGAN and BigGAN architectures at 64x64 resolution and BigGAN architecture for 128x128 resolution. However, they lack comparison with vanilla BigGAN (trained with hinge loss) and the explanation of the architectual differences. Authors only mention the lack of self-attention, yet the used architecture has much less parameters than the original BigGAN; it is also unclear how BigGAN for 64x64 resolution should look like. In fact, the choice of BigGAN for 64x64 resolution is questionable, as this model has been designed for datasets of much higher complexity.
4. In two out of three experiments, WBGAN-GP does not beat WGAN-GP. Given BigGAN's results at 64x64, it might be the case that bounding strategy just helps in the case of heavily-overparametrized model.
5. Definiton (8) is unclear: firstly, the objective L_theta is parametrized by parameters of discriminator (theta), over which the maximum on the rhs is taken (this follows in Appendices). Secondly, it has non-deterministic term d^lambda(..) which depends on random empirical distributions \hat(P_r), \hat(P_g) (in Proposition 8. which follows, authors take expectation over a Wasserstein distance between these distributions).
6. Remark 2's proof is incorrect. At the beginning of p.13 authors take D_theta(x) = sign(P_r(x) - P_g(x)), which is not a Lipschitz function. Also P_r and P_g may not be discrete so the entire expression does not make sense.
7. Remark 2 is not true non-deterministic formulation of (8). It is possible that P_g != P_r and \hat(P_g) = \hat(P_r); in such case sinkhorn distance is 0 and the remaining terms of rhs of (8) cancell out (e.g. P_r is a Dirac delta at 0 and P_g = Bernoulli(0.1), then it is possible that \hat(P_r) = \hat(P_g) = delta(0)). Perhaps re-shaping definition (8) so that it is deterministic would help achieving correctness of Remark 2.
8. Remark 3. is unclear: in what sense expression (8) can be optimized by gradient descent?
9. The paper seems rougly written (see below) and the text needs some polishing.

Typos/ requires clarification:
[p2.] Backgrounds -> background
[p.3] corresponds a \lambda -> corresponds to \lambda
[p.3] 'The proposed bounded strategy' -> bounding
[p.3] 'We name it general in two folds' (?)
[p.7] 'High-resolution' - 128x128 is not high.
[p.8] Discussions -> Discussion

**Experience Assessment:**

I have published in this field for several years.

**Review Assessment: Checking Correctness Of Derivations And Theory:**

I carefully checked the derivations and theory.

**Review Assessment: Checking Correctness Of Experiments:**

I carefully checked the experiments.

**Review Assessment: Thoroughness In Paper Reading:**

I read the paper thoroughly.

---

### Official Review · AnonReviewer3 · 2019-10-23
**Official Blind Review #3**

**Rating:** 3

**Review:**

This work represents some new effort to stabilize & improve the training of Wasserstein-based generative networks. To this end, the author(s) have proposed to combine both the primal and dual formulation of Wasserstein distance. By "clipping" the dual estimate wrt the primal estimate, or simply an arbitrarily set upper bound,  the author(s) hope to overcome the volatile training of WGANs which might be the consequence of an unbounded dual. Despite the fact that the overall idea is new and the results supported the claim, I am not convinced of its significance. It's an okay paper, but I am expecting more from an ICLR paper. As such, I am willing to endorse this submission. A more detailed review can be found below to help the author(s) improve their work.

Strength.
+ I like the idea of "bounding" the dual W-distance estimate, although, in theory, the observed degeneracy should not happen if the network is always properly regularized.
+ The overall writing is clear and well-motivated.

Weakness.
- The key observation that motivates this study is that W-distance often rises rapidly during the early training phase. This can be addressed by alternative tricks, say hot start the discriminator by pretraining it to be close to Lipschitz-1, or clipping the discriminator gradient to avoid drastic changes to the function. An ablation study is justified to pin down the real cause of this gain.
- It is not clear from the paper whether the  []_+ op back-propagates gradient or not, which will make a big difference here. If it backpropagates gradients, then when the dual estimate is larger than the threshold, then the computation is wasted (zero gradients). If not so, then the proposed solution does not affect gradient at all.
- It's worth noticing that vanilla Sinkhorn distance does not work well for generative purposes (see recent MMD-GAN and Sinkhorn-GAN papers, which all computes Sinkhorn distance in the feature space rather than data space). This questions the necessity of using the Sinkhorn estimate and why it works here.


**Experience Assessment:**

I have published one or two papers in this area.

**Review Assessment: Checking Correctness Of Derivations And Theory:**

I assessed the sensibility of the derivations and theory.

**Review Assessment: Checking Correctness Of Experiments:**

I assessed the sensibility of the experiments.

**Review Assessment: Thoroughness In Paper Reading:**

I read the paper at least twice and used my best judgement in assessing the paper.

---

### Official Review · AnonReviewer2 · 2019-10-24
**Official Blind Review #2**

**Rating:** 6

**Review:**

This paper proposes bounding the Wasserstein term in WGAN with an aim to stabilize training. The basic framework of the proposal, termed WBGAN, is presented in Section 3.1, and an instantiation using the Sinkhorn distance is described in Section 3.2.

The proposal is demonstrated to empirically improve stability over the baselines in the mid-resolution experiments presented in Section 4.2. On the other hand, in the high-resolution experiments in Section 4.3, the authors suggest that WBGAN with Sinkhorn does not scale well, and show some results with the bound determined on the basis of a separate run to investigate the converged value of the Wasserstein term, which is impractical. Moreover, the empirical performance seems comparable to the baselines in Figure 7. Because of these I would judge the contribution of this paper not strong, so that I would recommend weak acceptance.

In page 3, lines 20-21, the authors argue "a possible reason," which I do not understand. Take WGAN as an example. With weight clipping taking place over the whole training process, the Lipschitz constraint should automatically be satisfied even in the initial training phase, which would invalidate the authors' argument here.

In Section 3.2.1, I do not understand why the authors assume that the relevant distributions are in P_p(X) without specifying the value of p. Is it meant to be P_1(X)?

In Appendix A, I do not understand the sentence "Suppose \hat{\mathbb{P}}_{g_i} are the independent empirical measures drawn from \hat{\mathbb{P}}_g." It says as if samples from \hat{\mathbb{P}}_g were measures, but they are actually values in X.

Those figures showing the FID (Figures 1, 2, 5, and 7) are drawn versus the "epochs", whereas those showing the Wasserstein terms (Figure 4) and the generator loss (Figure 5) are drawn versus the iterations, which makes comparison between them impossible.

Page 1, line 43: which improve(s) the stability
Page 2, equation (3): Here a vector \Gamma 1_N is equated with an empirical distribution \hat{\mathbb{P}}, which does not make sense.
Page 2, line 40: where 1_N is a(n) N-dimensional
Page 3, line 8: Each \alpha corresponds (to) a \lambda
Page 3, line 21: As shown in Fig. (2->1)
Page 3, line 27: W should be italicized.
Page 3, line 42: Si(n)khorn distance
Page 6, caption of Figure 2: both each counterpart -> each of both counterparts
Page 6, line 1: discriminator -> critic
Page 6, line 24: both WGAN-div and WGAN-GPReal fail(s) to converge


**Experience Assessment:**

I have read many papers in this area.

**Review Assessment: Checking Correctness Of Derivations And Theory:**

I carefully checked the derivations and theory.

**Review Assessment: Checking Correctness Of Experiments:**

I assessed the sensibility of the experiments.

**Review Assessment: Thoroughness In Paper Reading:**

I read the paper at least twice and used my best judgement in assessing the paper.

---

### Decision · Program_Chairs · 2019-12-19

**Decision:**

Reject

**Comment:**

The paper presents a framework named Wasserstein-bounded GANs which generalizes WGAN. The paper shows that WBGAN can improve stability.

The reviewers raised several questions about the method and the experiments, but these were not addressed.

I encourage the authors to revise the draft and resubmit to a different venue.